# Registered report: Survey on attitudes and experiences regarding preregistration in psychological research

**Lisa Spitzer** *, Stefanie Mueller

Leibniz Institute for Psychology (ZPID), Trier, Germany

* ls@leibniz-psychology.org

## Abstract

### Background

Preregistration, the open science practice of specifying and registering details of a planned study prior to knowing the data, increases the transparency and reproducibility of research. Large-scale replication attempts for psychological results yielded shockingly low success rates and contributed to an increasing demand for open science practices among psychologists. However, preregistering one's studies is still not the norm in the field. Here, we conducted a study to explore possible reasons for this discrepancy.

### Methods

In a mixed-methods approach, we conducted an online survey assessing attitudes, motivations, and perceived obstacles with respect to preregistration. Respondents ($N = 289$) were psychological researchers that were recruited through their publications on Web of Science, PubMed, PSYNDEX, and PsycInfo, and preregistrations on OSF Registries. Based on the theory of planned behavior, we predicted that positive attitudes (moderated by the perceived importance of preregistration) as well as a favorable subjective norm and higher perceived behavioral control positively influence researchers' intention to preregister (directional hypothesis 1). Furthermore, we expected an influence of research experience on attitudes and perceived motivations and obstacles regarding preregistration (non-directional hypothesis 2). We analyzed these hypotheses with multiple regression models and included preregistration experience as a control variable.

### Results

Researchers' attitudes, subjective norms, perceived behavioral control, and the perceived importance of preregistration significantly predicted researchers' intention to use preregistration in the future (see hypothesis 1). Research experience influenced both researchers' attitudes and their perception of motivations to preregister, but not the perception of obstacles (see hypothesis 2). Descriptive reports on researchers' attitudes, motivations and obstacles regarding preregistration are provided.

This is a Registered Report and may have an associated publication; please check the article page on the journal site for any related articles.

**Data Availability Statement:** All data (including meta-data about variables and values and coded

comments for the qualitative analyses) are publicly accessible in the digital research repository PsychArchives (https://doi.org/10.23668/psycharchives.8400).

**Funding:** The authors received no specific funding for this work.

**Competing interests:** The authors have declared that no competing interests exist.

## Discussion

Many researchers had already preregistered and had a rather positive attitude toward preregistration. Nevertheless, several obstacles were identified that may be addressed to improve and foster preregistration.

## Introduction

Ever since Ioannidis [1] argued that most published research findings are false and a multitude of attempts failed to replicate previously significant effects (e.g., [2]), the reliability of published research findings has been a subject of discussion across many scientific disciplines. Summarizing these concerns, the term *replication crisis* [3] arose, where *replicability* refers to a study arriving at the same conclusion after collecting new data [4]. When *Nature* [5] conducted a survey of more than 1500 researchers of multiple disciplines, 70% of researchers reported that they had failed to replicate studies by others, and more than 50% had failed to replicate their own studies. Overall, 90% of surveyed researchers indicated their belief in a slight or even significant crisis [5]. In psychology, multiple large-scale research projects attempted to replicate significant effects published in top tier journals. Strikingly many attempts failed (i.e., they did not yield significant effects) as shown by replication rates varying between 36 [2] and 85% [6]. Among successfully replicated effects, the effect sizes were considerably smaller than originally reported.

It has been reasoned that false positives, i.e., effects that are significant in studies but do not exist in reality, contribute to low replicability [7]. The high rate of false positive research results has largely been attributed to "questionable research practices" (e.g., see [8, 9]), a collective term for any "exploitation of the gray area of acceptable practice . . . (which can) increase the likelihood of finding evidence in support of a hypothesis" ([9] p. 524). Examples for these practices are the failure to control for biases, selective reporting of significant results, *p*-hacking, or revising the hypotheses to match the results, also known as HARKing (see [8, 10–14] for details).

### Preregistration—on the rise?

The preregistration of studies has been proposed to detect and counter these questionable research practices (e.g., see [15], and see [10] for an overview of other open science techniques). A preregistration is a research plan that is time-stamped, created before the data has been collected or examined, and most often submitted to a public registry, thus making planned study details available to others (possibly after an embargo period) [15, 16]. If the research plan changes afterwards, either a new version needs to be added or the deviations will be apparent when comparing the preregistration to the final manuscript. Thus, preregistration aims for a transparent presentation of what was planned at a certain time point and what changes may have been made to a study until its publication. Evidence from other scientific disciplines indicates that preregistration indeed works, i.e., it increases the transparency of the research process, and reduces questionable research practices and the rate of false positive findings (e.g., see [17, 18] for examples from clinical trials and epidemiology). Correspondingly, a recent study showed that preregistration, among other open science techniques, can drastically increase the replication rate: In their study, Protzko and colleagues [19] tested the prospective replicability of 16 novel empirical findings, using preregistration among other

proposed current optimal practices. Here, 86% of effects could be replicated ($p < .05$) and effect sizes were 97% that of the original studies.

However, while preregistration is already well-established in other scientific disciplines and is mandated, for example, in medicine [20], it has been frequently demanded as a means to counter questionable research practices but is still not widely practiced in psychology.

On the one hand, in response to the replication crisis, many psychologists have committed themselves to the advancement and promotion of open science techniques such as preregistration (e.g., see [10, 13, 15, 21–23]). For example, Nosek et al. [21] describe preregistration as "hard, and worthwhile", while Wagenmakers and Dutilh [23] posit "seven selfish reasons for preregistration". Indeed, the number of preregistrations in psychology is increasing. For instance, the number of preregistrations on the Open Science Framework (OSF), a platform for sharing research materials, has been approximately doubling every year between 2012 and 2017 [22], and in a survey which was conducted in 2018, 44% of the sampled psychological researchers indicated having preregistered a hypothesis or analysis until 2017 [24].

Yet, looking at the fraction of published studies that were actually preregistered paints a different picture. In their recent study, Hardwicke et al. [25] found that only 3% of 188 examined articles from 2014 to 2017 which were randomly sampled from the literature included a preregistration statement, which contradicts the more positive outlook by [22, 24]. Stürmer et al. [26] also found that when early career researchers were asked about questionable research practices and open science, they deemed many open science practices necessary, yet toward preregistration they expressed more reluctance: Only about half of the participants found that preregistration was fairly necessary or very necessary, and even less indicated that they planned to consider preregistering their studies in the near future. Additionally, Logg and Dorison [27] showed that preregistration lags behind other open science practices such as open data and open material (as shown for articles in the journal "Organizational Behavior and Human Decision Processes").

A number of reservations are mentioned frequently when discussing preregistration in psychology, including the fear that it leaves no flexibility during study administration and eliminates the possibility to conduct exploratory analyses (as presented by e.g., [15, 16]). Some people are concerned that this might stifle discovery [28]. Others worry that someone might take their preregistered and thus publicly available study idea and publish it before them (so-called scooping, see [29]). Additionally, the time costs and effort are often seen as obstacles regarding preregistration (e.g., see [16]). Besides these worries, some authors also express an overall critique regarding the concept of preregistration. Szollosi et al. [30–32] argue that preregistration is redundant when good theories are tested and does not itself improve theories. Others argue that preregistration cannot prevent some questionable research practices [33] or might not fit well with all types of research (see [16]). Moreover, some studies found problems with the current implementation of preregistration such as poor disclosure of deviations from preregistered plans in finished manuscripts [34–37]. Although most of the listed arguments against preregistration are counter-argued by supporters of preregistration (e.g., that exploratory analyses are still possible [15, 16, 29]) and findings from other scientific disciplines underline its benefits (e.g., see [17–19]), these reservations persist, and some researchers remain skeptical.

## Aim of this survey

Previous surveys mainly inquired about preregistration in the more general context of open science [24, 26, 38, 39]. Meanwhile, comprehensive studies focusing on preregistration are lacking. We aimed to close this gap by exploring thoughts, motivations, and perceived

obstacles of psychological researchers toward preregistration and how these are influenced by the time someone has worked in research or actual experience with preregistration, that is, whether someone preregistered a study in the past. For instance, we wanted to explore the data to find out whether the low rate of preregistrations is caused by fear of the unknown or based on negative experiences (and which ones), as well as whether the increase of preregistrations is driven by a few active supporters while others reject or are indifferent toward it. Thus, we aimed to shed light on the outlined discrepancy of public support for preregistration on the one hand, and a low fraction of preregistrations on the other, while also identifying possible roadblocks for preregistration in psychology. Mixed-methods were used, including both qualitative and quantitative approaches.

Additionally, we investigated two specific research questions: First, we examined which factors facilitate or prevent preregistration (*research question 1*). The theory of planned behavior [40, 41] is a prolific, influential (e.g., see [42]) and well-researched (e.g., see [43–48]) psychological theory that aims to predict social behavior and has been applied across various contexts (e.g., health). According to this theory, the intention to perform a behavior can be seen as a direct antecedent of the actual behavior. In this framework, the intention to preregister predicts preregistration, and how the intention is formed may be informative for effectively promoting this behavior. To our knowledge, this has not yet been studied in the context of preregistration or open science. As described by Ajzen and colleagues [40, 41], three aspects influence intentions which are defined as follows: 1) Attitudes toward the behavior which result from the ratio of perceived advantages to disadvantages of performing the behavior, 2) the subjective norm which represents the perceived social pressure to perform or not perform the behavior, and 3) the perceived behavioral control which focuses on the question if the subject has the resources and skills to perform the behavior or not (also see [43–48] for meta-analytical support of this model, and [49] for an overview). We measured attitudes toward preregistration as well as subjective norms and perceived behavioral control through items in an online questionnaire and investigated how they influence researchers' intention to preregister their studies in the future. Based on the model's postulations, we expected that more favorable attitudes and subjective norms as well as higher perceived behavioral control positively influence the intention to use preregistration. As the relative importance of attitudes, subjective norms and perceived behavioral control differs in dependence of considered behaviors, situations, and populations [40, 41], we tested which of these is the strongest predictor for the intention to use preregistration. We also included the perceived importance of preregistration as moderator for the strength of the influence of attitudes on intention, and the preregistration experience as a control variable. Such an extension of the model made it possible to compensate for potential non-attitudes (e.g., see [50]) and a potential sampling bias toward researchers that have already preregistered, and is explicitly allowed by the theory of planned behavior [40, 49].

Regarding the intention formation, we formulated the following hypotheses:

1. The theory of planned behavior [40, 41] can be applied to the context of preregistration to significantly predict researchers' intention to preregister their studies in the near future, using a moderated multiple regression model. We predicted that:

   1.1 More beneficial attitudes are a positive predictor for the intention to preregister.

   1.2 The perceived importance of preregistration moderates the effect of attitudes on intention positively.

   1.3 The perceived importance of preregistration is a positive predictor for the intention to preregister.

1.4 Beneficial subjective norms are a positive predictor for the intention to preregister.

1.5 Higher perceived behavioral control is also expected to be a positive predictor.

1.6 These predictors combined can significantly predict researchers' intention to preregister.

Second, we examined whether research experience predicts attitudes and the perceived intensity of motivations and obstacles (*research question 2*). Research experience was operationalized as the number of years someone indicates they have worked in psychological research. Early career researchers are oftentimes seen as the driving force of the open science movement (e.g., see [51]). We investigated if the research experience indeed has an influence on researchers' responses about preregistration (a similar effect was reported by Abele-Brehm et al. in a comparison of academic groups regarding hopes and fears toward open science [38] and by Logg and Dorison when inspecting generational differences [27]).

Regarding this second research question, the following hypotheses were tested:

2. We predicted that the research experience, that is, the amount of time someone has already worked in psychological research, has an influence on attitudes, motivation, and perceived obstacles regarding preregistration. Specifically, we conducted three multiple regressions (including preregistration experience as a control variable), and we posited the following non-directional hypotheses:

2.1 Research experience is a predictor for attitudes regarding preregistration.

2.2 Research experience is a predictor for the strength of motivation to preregister.

2.3 Research experience is a predictor for how strongly obstacles to preregister are perceived.

The present survey aimed to sample the general population of psychological researchers by recruiting participants whose articles appear on Web of Science, PubMed, PSYNDEX, and PsycInfo, as well as the subgroup of researchers who have preregistered before and who were identified through their preregistrations on OSF Registries.

## Methods

The methods and analyses reported below were implemented as described in the Registered Report Protocol (DOI: 10.1371/journal.pone.0253950). For a transparent overview, all deviations are summarized in Table 1.

### Material

The online survey was created with the software SoSci Survey (version 3.2.29) [52] and was supplied via www.soscisurvey.de. It was presented in English. Items of the survey could be categorized in nine categories: 1) Sociodemographic questions, 2) items concerning the general usage of preregistration, 3) an *attitude* scale consisting of 24 items indicating the overall attitudes of participants regarding the concept of preregistration, 4) a *subjective norm* scale of eight items representing perceived social norms and pressure, 5) a *perceived behavioral control* scale of five items inquiring about researchers' perceived control over the potential preregistration of their studies, 6) an *intention* scale of three items inquiring about researchers' intention to use preregistration in the future, 7) items about motivations (a *motivation* scale including ten items measuring how strongly participants agree with potential motivations to preregister, plus additional more open items), 8) items about perceived obstacles (an *obstacle* scale featuring ten items measuring how strongly participants agree with potential obstacles to preregister, plus additional more open items), and 9) various open questions inquiring about suggestions

**Table 1. Deviations from the preregistered procedure.**

| Section | Description and justification |
|---------|------------------------------|
| Introduction & Discussion | New studies were added to the *introduction* and *discussion* sections that have been published while we prepared the second part of this Registered Report. |
| Methods | The section *sampling procedure* was moved down in the manuscript and divided into smaller subsections to facilitate readability of the article. In its current form, everything that was taken directly from the descriptions of the Registered Report Protocol is displayed in the beginning of the manuscript, and everything that was added (results, information about the data collection, e.g., proportion of participants recruited through each database etc.) is displayed in the second part of the manuscript. |
| Pilot study | Reliability analyses for the pilot study were slightly adjusted for the Registered Report compared to the Protocol because an error was discovered in the analysis script for the pilot recoding data. This did not change the interpretation or implications of the pilot study data. |
| Exclusion, missing data, and sample size | In the Registered Report Protocol, it was not specifically defined that participants who expressed unfaithful participation would be excluded for the descriptive reports, however, these were excluded ($n = 9$). <br> In addition to the preregistered exclusion criteria, two participants were excluded since due to a technical error, no informed consent was obtained. <br> Due to an error, the maximum quota size for the "doctoral degree" quota was exceeded by one participant. <br> Due to exclusions after data collection (which were applied as specified in the Registered Report Protocol), the target sample size of $N = 296$ was missed by seven participants. |
| Data analysis and preprocessing | The list of used R packages was updated in the manuscript based on which packages were used in the main study analyses. <br> It was planned to insert the results calculated in R into the manuscript using StatTag, however, this was not possible because StatTag was not available for the used R version. |
| Descriptive reports | Participants' locations were displayed as continents, not as countries, for reasons of brevity and anonymity. <br> For brevity and comprehensibility, the descriptive reports are displayed in plots or only in the text instead of providing frequency tables. Furthermore, the reports of the open text input items include only those themes that were mentioned in more than 5% of the respective comments; and for the "other" comments, themes which were only indicated by single participants are omitted (however, all data and coded themes are available in PsychArchives). <br> It was not clearly defined in the Registered Report Protocol that the ANOVAs to compare intention/importance between academic groups would be conducted separately for participants with/without preregistration experience, however, this was done in the Registered Report. Additionally, a Brown-Forsythe test for equality of means was used for one of these because the assumptions for the ANOVA were not fulfilled. |
| Analysis Scripts | The analysis scripts were edited slightly to enable correct functioning, improve the code (e.g., by using *tidyverse* instead of Base R), create plots, etc. No changes were made to the code for analyzing the hypotheses. <br> All changes from the preliminary scripts included in the Registered Report Protocol are documented at the end of the respective analysis script. |

for improving preregistration. Some items were adapted from other surveys that focused on related topics [26, 38] and were complemented with additional items generated in reference to other theoretical and empirical works on preregistration, open science [33, 53–55], and the theory of planned behavior [40, 41, 49, 56]. These sources were selected based on their theoretical relevance to the topic at hand (i.e., open science practices and theory of planned behavior). The identified items were either adapted (i.e., wording was changed slightly to make them easier to understand or make them fit the present topic, e.g., "I have more trust in research findings when the study has been preregistered." instead of ". . . when the respective data are published." [38]) or new items were generated using the concepts presented in the literature based on face validity. The items of the present survey as well as the original items including references are available in the supporting information (see S1 Table). For most of the original items, no validity measures are given in the literature. Only for the theory of planned behavior, predictive validity is considered in more detail [49]. However, since none of the survey items were used in their original form but were adapted or newly created based on the given concepts, no validity measures can be provided. Instead, only face validity is assumed. This validity evaluation is based on Flake and Fried [57].

Different item formats were included in the survey. All scale items (attitudes, subjective norm, perceived behavioral control, intention, motivations, and obstacles) were answered with a seven-point labeled answer scale as recommended by Ajzen [40] (1 = "Strongly disagree", 2 =

"Disagree", 3 = "Slightly disagree", 4 = "Neither agree nor disagree", 5 = "Slightly agree", 6 = "Agree", 7 = "Strongly agree"). Scales were recoded from "1 to 7" to "-3 to +3" for data analyses yielding a middle category which has absolute meaning (i.e., 0 = neutral opinion, neither agreement nor disagreement). For the statistical analyses, the mean scores of the scales were used to measure how participants perceived 1) preregistration (attitude scale), 2) subjective norms regarding preregistration (subjective norm scale), 3) their own control about using pre-registration or not (behavioral control scale), 4) their intention to use preregistration in the future (intention scale), and 5) their motivations (motivation scale) and 6) obstacles to prereg-ister (obstacle scale). For each participant, the mean for each scale was calculated and used as the score.

Other items included a single or multiple choice response format, or the option for open text input. Whenever applicable, response options were displayed in randomized order to eliminate potential sequence effects.

## Procedure

Participants received the link to the survey via personal email or social media call (see section *sampling procedure*). After the welcoming page, participation information was displayed, and informed consent needed to be provided to proceed. Furthermore, a captcha (arithmetic task) needed to be completed as a safeguard against bot responses. Then, participants completed the main body of the survey which successively focused on the different item categories (sociode-mographic questions, general usage of preregistration, attitudes, subjective norms, perceived behavioral control, intention, motivations, perceived obstacles, and suggestions for improve-ment). Items which were relevant for the study procedure (e.g., informed consent, usability), as well as those important for the exclusions, quota sampling, and hypotheses tests were man-datory. Meanwhile, the items used for the descriptive reports were optional. Before any items related to preregistration were shown, a definition of preregistration was presented (see S1 and S2 Videos and S1 File) and correct understanding was checked, to ensure that all participants answered the items with the same concept in mind. At the end of the survey, participants had the option to participate in a lottery for 40 gift cards worth 50 € each and to sign up for receiv-ing a preprint of the survey results by entering their email address, which was saved separately from the other data. Lastly, a debriefing followed that stated the aim of the survey and the research questions that were investigated. It took participants about 18 minutes (*SD* = 8 min, *range* = 44 min) on average to complete the survey (times were adjusted for interruptions by replacing the completion time of pages that took participants more than two hours or 3 x *SD* of a normal distribution, with the page median of the other participants). Screen recordings of the survey's procedure are available in the supporting information (see S1 and S1 Videos). Additionally, a PDF of the questionnaire can be found in S1 File. After data collection, the win-ners were drawn, and the vouchers were awarded. The survey was approved by the ethics com-mittee of Trier University, Germany (approval number: 07/2020). This form of consent was obtained in writing.

## Pilot study

We conducted a pilot study to estimate the response rate and to test the recruitment method as well as the survey items. The pilot study featured mostly the same items as the main study but additional items, prompting participants for feedback about comprehensibility via open text input fields, were presented ("Did you have any problems answering the items of this page? Was anything unclear?"). Some items were adjusted based on the results.

We invited $N = 200$ participants to partake in the pilot study (100 were invited from OSF Registries, 50 from Web of Science, and 50 from PubMed) and sent a reminder one week after the initial invitation. Participations typically occurred shortly after each email contact. The pilot study was accessible for one month overall. In this time, 29 participants (17 PhD students, three postdocs, seven professors, and two members of other academic groups; the latter were screened out) started the survey of which 20 completed it (14 PhD students and six professors), yielding an overall response rate of 10%. Out of these 20, 18 (90%) participants had preregistration experience. No specific patterns in dropout behavior were found (e.g., dropping out at a certain position). No floor or ceiling effects were found. Reliability analyses for the measured scales were conducted, which showed an excellent reliability for the attitude ($\alpha = .93$) scale, a good reliability for the motivation scale ($\alpha = .8$), an adequate reliability for the obstacle ($\alpha = .79$), and a moderate reliability for the subjective norm ($\alpha = .65$) and perceived behavioral control scale ($\alpha = .66$). One item of each scale correlated negatively with the remaining scale items (A11, M2, O8, SN2, and PBC3, see S1 Table). In the case of the PBC scale, Cronbach's alpha was over .10 higher if the respective item was excluded. Nevertheless, the items remained in the survey and were checked again in the main study, since the pilot results were based on a very small sample and could only be interpreted with caution. And indeed, the reliability for the main study sample was adequate to excellent, the items that were unreliable in the pilot study exhibited normal patterns and were included in the main analyses based on the specified criteria (see section *data analysis and preprocessing*). Reliability analyses for the pilot study were slightly adjusted for the Registered Report compared to its Protocol because an error was discovered in the analysis script for the pilot recoding data.

After conducting the pilot study, the sampling strategy was revised by extending the recruitment of participants to four instead of two general databases for psychological articles and by inviting less participants via OSF Registries (10% instead of 50%). Moreover, some comments indicated ambiguous interpretations which led us to slightly modify the wording of respective items or to change categories (i.e., academic position was replaced by academic degree in the sociodemographic part of the survey). Lastly, a few items were added, e.g., two items assessing intention, following a manual about constructing questionnaires based on the theory of planned behavior [56].

## Sampling procedure

**Power analyses.** Data from psychological researchers at different career stages were collected. The optimal sample size of $N = 296$ was determined by using G*Power [58, 59] in combination with a thorough review of the existing literature as described in the following paragraphs. Each of the power analyses described below was specified to achieve a statistical power of 95% at a given significance threshold of 5% ($\alpha = \beta = .05$). All power analyses are displayed in Fig 1 and are also included in the supporting information (see S2 Text).

To test which factors influence the intention to preregister (scale of three items, see *hypothesis 1*), a moderated multiple regression model was computed based on the rationale of the theory of planned behavior, which included six predictors: Attitudes (scale of 24 items), perceived importance of preregistration (one item in our questionnaire), attitudes x importance, subjective norm (scale of eight items), perceived behavioral control (scale of five items) [40, 41, 49], and preregistration experience (one item). The theory of planned behavior has been examined using meta-analytical approaches in various contexts (e.g., health behavior). The percentage of variance of intention that was explained by attitudes, subjective norm and perceived behavioral control combined, ranged between $30.4\% < R^2 < 44.3\%$ [43, 45–48]. We chose the lowest reported effect size ($R^2 = 30.4\%$) as minimal effect size of interest. The power analysis for the

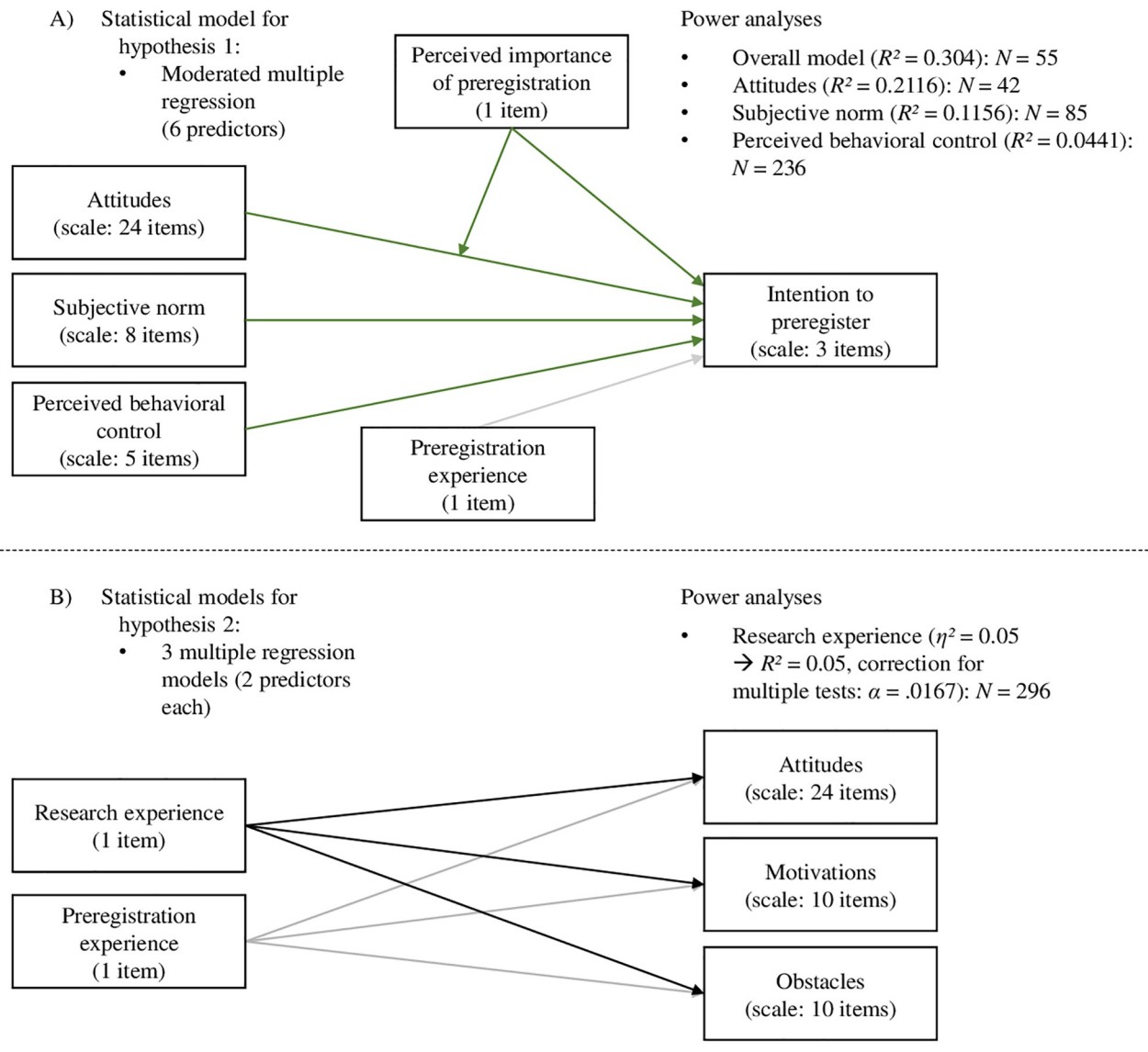

**Fig 1. Overview of the power analyses.** The detailed statistical models and corresponding power analyses are displayed for A) the moderated multiple regression that were computed for testing hypothesis 1 and B) the multiple regressions that were used to test hypothesis 2.

overall regression model yielded an optimal sample size of $N = 55$ to be able to detect the determined effect size given the effect exists (see *hypothesis 1.6*). Additional power analyses were conducted to compute the optimal sample size to test the individual predictors. As comparable effect sizes, $R^2$ based on the averaged correlations of individual variables were searched for in the aforementioned meta-analyses, and the smallest ones were chosen for each power analysis. This resulted in an optimal sample size of $N = 42$ for testing attitudes (see *hypothesis 1.1*), $N = 85$ for testing subjective norms (see *hypothesis 1.4*), and $N = 236$ for testing perceived behavioral control (see *hypothesis 1.5*) as predictors for intention (with $\alpha = \beta = .05$). Since perceived importance and its interaction with attitudes, as well as preregistration experience were not originally included in the model but added by us, no comparable effect sizes were available and thus, no power analyses were conducted for these variables.

To investigate whether research experience has an effect on attitudes (see *hypothesis 2.1*), and the perceived intensity of motivations (see *hypothesis 2.2*) and obstacles (see *hypothesis 2.3*) regarding preregistration, three regression models were computed, again including preregistration experience as a control variable. As comparable effect size, the smallest reported effect size for group differences by Abele-Brehm et al. [38] was used, and the corresponding *F* value was used to calculate $R^2$ for the power analyses. For these regression models, the alpha level was corrected using the Bonferroni-Holm method to account for multiple testing. Thus, the power analyses were calculated for the smallest alpha of 1.67% (5% divided by 3). These power analyses suggested an optimal sample size of *N* = 296 to test the influence of research experience on attitudes (see *hypothesis 2.1*), perceived intensity of motivations (see *hypothesis 2.2*), and perceived intensity of obstacles (see *hypothesis 2.3*) regarding preregistration.

As the optimal sample size for the regression models to test hypothesis 2 was the highest computed necessary *N*, it constituted the targeted sample size of *N* = 296.

**Academic quotas.** To ensure that our sample contained researchers with varying research experience, a quota sampling was used to evenly collect data from researchers with different degrees, that is, bachelor's degree, master's degree, doctoral degree, or habilitation and/or full professorship. Specifically, a quota sampling was used with a 25% quota for each subgroup. Yet, as it was anticipated that it might be the case that some quotas might not be filled, quotas were defined for a slightly bigger sample than the necessary *N* = 296 to compensate for this eventuality. In particular, *N* = 400 was the basis for our quotas which were therefore *n* = 100 for each group. This enabled us to reach the a priori computed power even if not all quotas could be filled, while still ensuring that the potential overrepresentation of individual groups remained within reasonable limits.

By using this quota sampling, we were able to reach individuals from all academic groups, but persons with a master's degree (*n* = 92) or doctoral degree (*n* = 101) were more strongly represented than those with a bachelor's degree (*n* = 38) or professors (*n* = 58). Due to an error, the maximum quota size for the "doctoral degree" quota was exceeded by one participant. When incomplete data sets were also considered (as was the case for the descriptive reports, see section *exclusion*, *missing data*, *and sample size*), the groups were substantially larger (bachelor's degree: *n* = 63; master's degree: *n* = 124; doctoral degree: *n* = 161; habilitation and/or full professorship: *n* = 72).

**Recruitment through different databases, email lists, and social media.** To recruit the sample, a method similar to the one by Field et al. [53] was used. The term "psychology" was searched for on specified databases, and resulting hits were sorted from newest to oldest. These documents were scanned for authors whose email addresses could be found via institutional or personal websites linked to the respective work, or by searching for the author via Google, Google Scholar, and ResearchGate. Duplicated email addresses that were sampled from different databases were excluded. Identified persons were invited to participate in the survey via email.

For this search, the databases Web of Science (https://apps.webofknowledge.com/), PubMed (https://pubmed.ncbi.nlm.nih.gov/), PSYNDEX (https://www.psyndex.de/), and PsycInfo (https://www.apa.org/pubs/databases/psycinfo) were used to recruit a representative sample of psychological researchers.

Based on the results by Hardwicke et al. [25], we anticipated that only a small proportion of the research articles found on the general databases would include a preregistration statement, which could have led to a small number of participants with any preregistration experience. Thus, we decided to send 10% of invitations to authors of preregistrations in order to ensure that our survey would also include a sufficient number of participants who had preregistered before. These participants were identified via the preregistration platform OSF Registries

(https://osf.io/registries). As this may have introduced a sampling bias towards researchers who lean more positive towards preregistration, preregistration experience was controlled for in the statistical analyses aimed to draw inferences about the general population of psychological researchers by including it as a control variable in the hypotheses tests, and by providing descriptive reports separately for participants with and without preregistration experience.

A customized link was distributed among participants of each database so that the recruitment source could be inferred. Overall, 17.37% of participants were recruited through PSYNDEX, 10.17% through OSF, 9.75% through Web of Science, 8.47% through PsycInfo, and 6.57% through PubMed. Additional 23.73% of our sample's participants were recruited through email lists and 23.94% through social media. When only taking complete datasets into account, 21.8% of participants were recruited through PSYNDEX, 14.88% through OSF, 11.07% through Web of Science, 6.57% through PubMed, 6.23% through PsycInfo, 21.8% through email lists, and 17.65% through social media.

For all sources, around half or more participants indicated having preregistration experience (OSF: 97.67%; PsycInfo: 68%; PSYNDEX: 67.16%; email lists: 61.97%; social media: 61.76%; PubMed: 60.87%; Web of Science: 47.06%). The database participants were recruited from was considered in the analyses as described in the section *descriptive reports*.

**Response rates and invitation procedure.** Response rates were rather low in many studies that relied on researchers as sample (e.g., [9, 26, 38, 39, 53, 54, 60]). Based on these studies and based on the insights from our pilot study (see section *pilot study*), we anticipated a response rate of around 10%. To compensate for this, we invited $N = 2960$ persons ($n = 666$ via Web of Science (22.5%), $n = 666$ via PubMed (22.5%), $n = 666$ via PSYNDEX (22.5%), $n = 666$ via PsycInfo (22.5%), and $n = 296$ via OSF Registries (10%)) to reach our target sample size of $N = 296$ (10% of 2960).

Since the first invitation wave did not result in a sufficient sample size, an additional invitation wave was conducted, as pre-specified in the Registered Report Protocol (DOI: 10.1371/journal.pone.0253950). In this second wave, participants were subsequently invited for quotas that had not yet been filled. For each open quota, the response rate of the first invitation wave was used to calculate how many more participants needed to be invited to fill the quota. This procedure is described in more detail in the supporting information (see S3 Text). Individuals of the first wave had approximately one month and one week overall to participate in the survey, individuals of the second wave had two weeks to participate. Both groups received a reminder email one week after invitation.

Of all participants who were included in the analyses (see section *exclusion*, *missing data*, *and sample size)*, $n = 247$ were recruited though this predefined database recruitment strategy, from which $n = 175$ completed the survey. This corresponds to an actual response rate of 8.34% for our database recruitment (5.91% for complete datasets).

Additionally, the survey was advertised on social media (Facebook, Twitter), and via student and researcher specific mailing lists, which were seen as additional sources of participants. Hereby, $n = 225$ additional participants ($n = 114$ complete datasets) were recruited.

During data collection, the targeted sample size was originally surpassed by one dataset, but due to the exclusion of inappropriate responses after data collection, it was missed by seven records (i.e., $N = 289$ participants were included in the hypotheses tests, see section *exclusion*, *missing data*, *and sample size)*.

**Timeline data collection.** The collection of email addresses started on January 4, 2021. The first wave of invitations was sent on November 15, 2021. Overall, $N = 1615$ persons clicked on the survey link and $N = 513$ started the survey. The planned sample size was reached just before the end of the set time frame, (however, some data were excluded post-data collection, see section *exclusion*, *missing data*, *and sample size)*. As preregistered, the survey was still

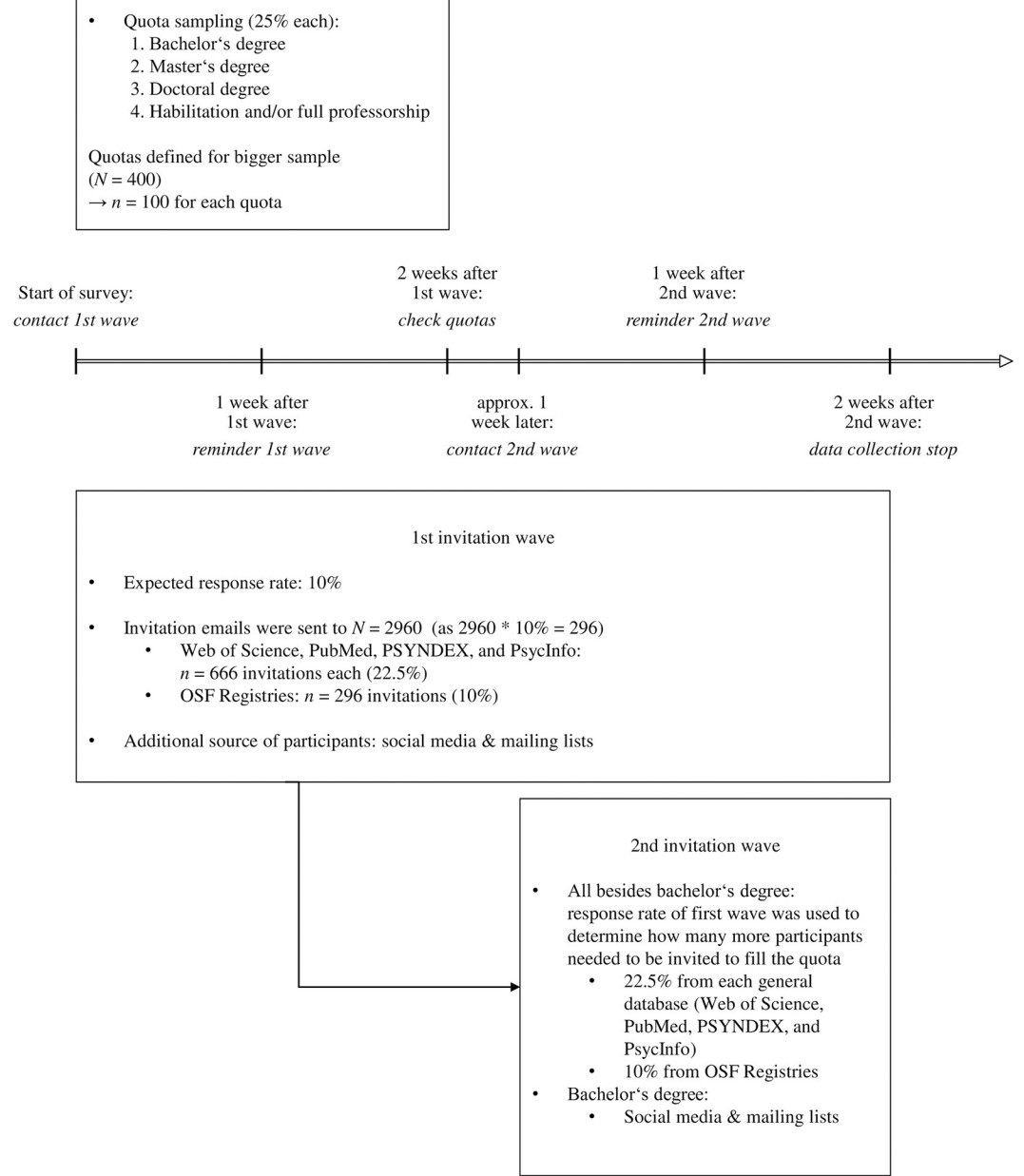

**Fig 2. Survey timeline and corresponding actions.** The arrow illustrates the timeline while the boxes contain details about the sampling procedure.

accessible to participants that were already invited until the end of the defined timeframe, but recruitment was discontinued. Data collection was stopped based on the set criteria on December 21, 2021. The timeline of the data collection is displayed in Fig 2.

**Exclusion, missing data, and sample size.** All participants that indicated that their research or studies did not fall within the scope of psychology ($n = 15$) or did not have at least a bachelor's degree in psychology, and thus, could not be assigned to the respective quota

($n$ = 7), were screened out at the beginning of the survey. Thus, they were directed to an exit page rather than to the main body of the survey, were not counted into the quotas, and were not considered for the data analyses as the survey targeted a sample from research-oriented psychology.

At the end of the survey, participants were asked whether they responded faithfully. Here, nine participants indicated that their data should not be used in the analyses. Only data of participants that indicated having at least a bachelor's degree in psychology, indicated faithful participation, and completed all pages were counted for quota fulfillment. By screening the open text inputs, it was additionally checked if participants included any inappropriate responses like advertising or offensive comments. Those participants ($n$ = 8) were excluded from all analyses. In addition to these preregistered exclusion criteria, two participants were excluded since—due to a technical error—no informed consent was obtained.

Overall, this yielded a sample of 289 valid participants who completed all pages of the survey and, additionally, 183 valid participants who started the survey without completing it. The 289 complete datasets (42.21% male, 52.94% female, 1.04% other, 1.38% preferred not to answer, 2.42% did not respond; $Mean_{age}$ = 34.99 years, $SD_{age}$ = 10.72 years; 70.59% from Europe, 23.18% from North America, 2.08% from Asia, 2.08% from Australia, 0.35% from Africa, 0.35% from South America, 1.38% did not respond) were analyzed when testing the planned hypotheses. The 289 complete datasets together with the 183 incomplete datasets ($N$ = 472) were used for calculating descriptive statistics where applicable (37.08% male, 47.03% female, 0.64% other, 1.69% preferred not to answer, 13.56% did not respond; $Mean_{age}$ = 35.08 years, $SD_{age}$ = 11.07 years; 58.26% from Europe, 23.31% from North America, 2.97% from Asia, 1.69% from Australia, 0.64% from Africa, 0.64% from South America, 12.5% did not respond; participants' research topics are summarized in S4 Text). Sensitivity analyses showed that the exclusion of incomplete datasets for the hypotheses tests did not affect the results.

## Data analysis and preprocessing

This Registered Report was written in Microsoft Word and *R Markdown* (version 2.7) [61, 62]. R (version 4.1.1) [63] and the R-packages *afex* (version 0.27–2) [64], *Amelia* (version 1.7.6) [65], *car* (version 3.0–8) [66, 67], *corrplot* (version 0.84) [68], *lm.beta* (version 1.5–1) [69], *olsrr* (version 0.5.3) [70], *onewaytests* (version 2.6) [71], *psych* (version 1.9.12.31) [72], *stats* (version 4.4.1) [63], *tidyverse* (version 1.3.0) [73], and *writexl* (version 1.3) [74] were used for the analyses. Scripts for preprocessing, assumption testing and all analyses are available in the supporting information (see S1–S4 Scripts). Additionally, all data (including meta-data about variables and values) are publicly accessible in the digital research repository PsychArchives (https://doi.org/10.23668/psycharchives.8400).

Data were processed in the following way: Responses from all scale items were recoded as described in the section *material*, and the polarity of negatively poled items was reversed. Additionally, multiple choice questions were recoded (originally: 1 = "not selected" and 2 = "selected"; new: 0 = "not selected" and 1 = "selected"), and single choice items were transformed into factors. Comments were inspected. Furthermore, data was screened for any inappropriate responses (i.e., advertising, offenses) which led to an overall exclusion (see section *exclusion, missing data, and sample size*).

As a quality check, the attitude, subjective norm, perceived behavioral control, intention, motivation, and obstacle items were inspected with respect to floor or ceiling effects which would then be excluded from the analyses (specifically, items for which $\geq$ 90% of participants indicated the lowest or highest category). Additionally, Cronbach's alpha was calculated as an

indicator for reliability for each scale. We planned to exclude items that reduced the Cronbach's alpha of a scale by more than .10. Considering these three quality checks in the complete datasets, no survey items were excluded. Overall, reliability analyses of the survey items showed excellent reliability for the attitude ($\alpha$ = .94) and intention scale ($\alpha$ = .93), good reliability for the motivation ($\alpha$ = .8) and obstacle scale ($\alpha$ = .84), and adequate reliability for the subjective norm ($\alpha$ = .72) and perceived behavioral control scale ($\alpha$ = .72). Thus, all scale items were used to calculate mean scores for the attitude, subjective norm, perceived behavioral control, intention, motivation, and obstacle scale respectively for each participant. All preprocessing scripts are available in the supporting information (see S1 Script).

## Results

Below, we describe all preregistered descriptive reports and hypotheses tests. In addition to our planned analyses, various exploratory analyses were conducted, all of which are documented and described in the supporting information (see S4 Script and S5 Text).

### Descriptive reports

Several descriptive reports are provided to gain qualitative insight into the psychological research community's experiences, opinions, and behavior with respect to preregistration, and display participants' suggestions for improving the preregistration process. Since incomplete datasets were also used for these reports, parameters are based on all responses given to the respective item ($N$ is provided in each case).

Open text inputs of both the open "other" options of selection items and the open text input items were analyzed to identify common themes. The following mixed-methods approach was used: Two coders qualitatively identified themes mentioned by the participants and subsequently categorized all responses accordingly in order to receive a frequency measure that was analyzed quantitatively. Specifically, open text inputs of each item were shuffled and the first 10% of these shuffled responses were used to establish initial categories of themes. Coders read the responses and added each theme as a column in a coding sheet. It was coded whether the respective theme appeared in the other responses (= 1) or not (= 0). If a coder encountered new relevant themes, they were added and coded later. Nonsense responses were excluded during coding. Ambiguities were discussed and solved in pairs. If no solution could be found, a third coder was consulted. After all responses had been coded, the sums for each column (= theme) were calculated to obtain the frequency of how often a theme was mentioned over all responses. Results from the open text input items are displayed in the text and stacked bar plots, and the results of the analysis of "other" options were added to the individual items' response presentations. For brevity, the reports of the open text input items include only those themes that were mentioned in more than 5% of the respective comments, and for the "other" comments, themes which were indicated only by single participants are omitted. Wherever applicable, the results were compared between participants that had preregistered before or not. Both the coding sheets and the open text inputs are published alongside the other data. Furthermore, all analysis scripts are included in the supporting information (see S3 Script).

**Perceived importance and intention to preregister.** First, we compared the perceived importance of preregistration (seven-point labeled answer scale item) and the intention to preregister in the future (scale of three items) between researchers with different academic degrees by conducting one-factorial ANOVAs with intention/importance as the dependent variable and degree as the independent variable. These analyses were conducted separately for participants with and without preregistration experience and corrected for multiple testing using the

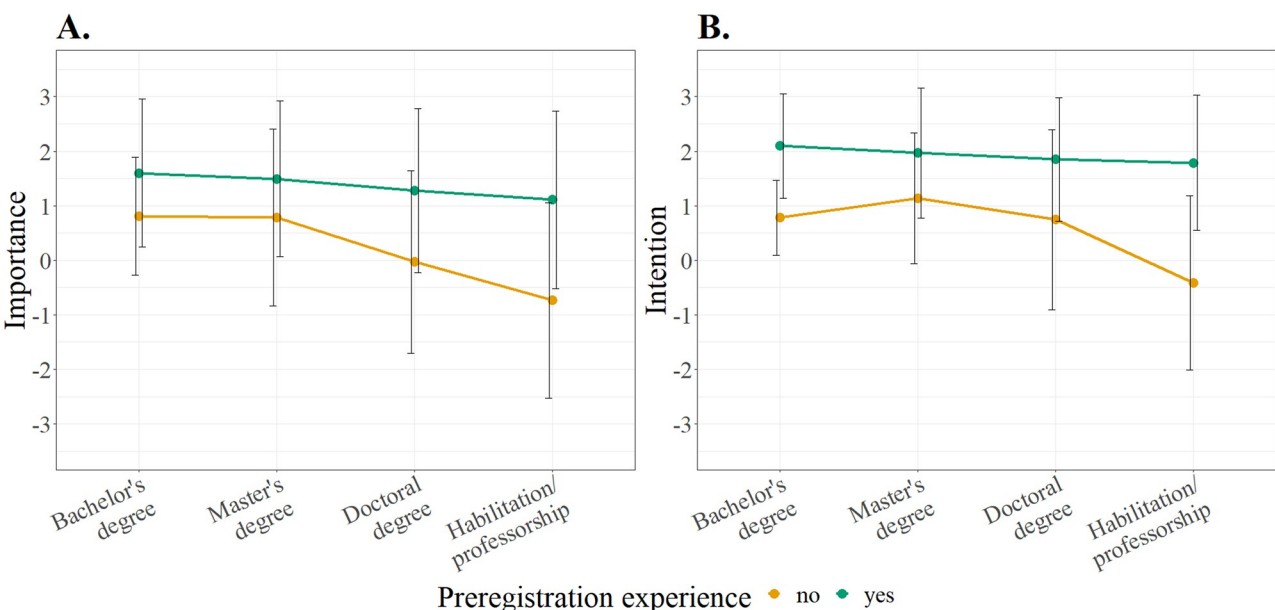

**Fig 3. Importance and intention.** Mean values for A) the perceived importance of preregistration (one item) and B) participants' intention to preregister (scale of three items, see S1 Table). Scales ranged from 1 = "Strongly disagree" to 7 = "Strongly agree" and were recoded to "-3 to +3" for data analyses. The error bars show + / − 1 *SD*.

Bonferroni-Holm method. For the analysis of intention in participants without preregistration experience, a Brown-Forsythe test for equality of means was used since variances were not equal and group sizes differed. The results showed that intention and perceived importance were similar between academic groups for participants with preregistration experience (both $p_{cor}$ = .952) but differed for participants that had not preregistered before (importance: $F(3, 96)$ = 4.38, $p_{cor}$ = .018; intention: $F(3, 63.35)$ = 4.94, $p_{cor}$ = .015), as displayed in Fig 3.

**Proportion of participants with preregistration experience.** To gain insight into how common preregistration is in the general population of psychological researchers, we report the mean proportion of participants who used preregistration in the past and how many pre-registrations they created on average, across the sample obtained from the article databases, that is, excluding participants recruited from OSF Registries. In this general sample ($n$ = 288 responses), 61.81% of researchers indicated having used preregistration in the past (47.83% of participants with a bachelor's degree, 60.92% of participants with a master's degree, 63.16% of participants with a doctoral degree, and 71.67% of participants with a habilitation or full professorship), having preregistered six studies on average (*Mean* = 6.05, *Median* = 3, *SD* = 10.68, *IQR* = 3, *range* = 80). When only the sample recruited through the databases is considered (thus excluding participants recruited via email lists and social media in addition to OSF Registries), the proportion of participants with preregistration experience remains almost unchanged (61.74% of 149 responses). Of the OSF sample, 97.67% of participants indicated having preregistration experience, while one participant responded that they did not preregister before (i.e., maybe a secondary author who was not actively involved in the preregistration).

**First contact with preregistration.** Moreover, also including the sample recruited from OSF Registries, participants' responses to various questions regarding participants' experiences with preregistration were investigated.

The majority of participants indicated that they first learned about preregistration in informal conversations with colleagues or peers (55.62% of 329 responses, multiple choice item). Around a quarter of participants furthermore indicated hearing about preregistration for the first time at official events at the workplace (27.36%), lectures at university (25.53%), or from their supervisors (24.32%), and some participants also learned about it through projects at their university (19.15%). In the "other" option, participants indicated that their first contact with preregistration was through journal articles (6.69%), social media (4.26%), conferences (2.43%), workshops (2.13%), the scientific community (1.23%), or editorial policies (0.61%). Meanwhile, 3.95% of participants indicated that they had not heard about preregistration before this survey.

Most participants indicated having created their first preregistration self-motivated (48.72% of 195 responses, multiple choice item). Other common reasons included informal conversations with colleagues or peers (34.36%), or it being suggested by their supervisor (29.23%) or co-authors (21.03%). In some cases, participants also created their first preregistration because it was mandatory for a project (12.82%) or to get funding (4.62%). Other reasons provided in the open text input section of this item were that it was considered useful or essential for publication (2.05%) and that it was part of courses that participants had attended (1.03%).

**Perceived benefits and drawbacks of preregistration.**   Reported benefits of preregistration (open text input item) were related to transparency (37.89% of 190 responses), trustworthiness of science (37.37%), or reduced uncertainty (5.26%). However, most frequently mentioned was that preregistration was helpful for study planning (41.58%). When asked what the long-term positive consequences of mandatory preregistration would be (open text input item), the vast majority also indicated transparency (60.31% of 257 responses). Less publication bias (28.79%), higher quality of research (27.24%), detailed study planning (24.51%), and replicability (12.45%) were other common themes.

When participants who had preregistered before were asked if their motivation to preregister had increased or decreased over time, more participants reported increased ($n = 67$) than decreased motivation ($n = 15$). The main reasons for a higher motivation (open text input item) were the feeling of preregistration becoming the norm (28.36% of 67 responses), that it improved their study structure (23.88%), the overall benefits of preregistration (19.4%), or increased knowledge about it (14.93%). Other reasons included transparency, that it allowed participants to clarify their own ideas, or that the preregistration process was simple (7.46% each).

Meanwhile, reported drawbacks of preregistration (open text input item) were, above all, its time costs (43.41% of 182 responses). Other disadvantages, though less commonly perceived, comprised loss of flexibility (17.58%), increased effort (15.38%), perceived devaluation of exploratory research (7.14%), perceived incompetence if deviations occur (6.59%), it not being suitable for all projects (6.59%), or the possibility of scooping (5.49%). Time costs (28.22% of 241 responses), reduced flexibility (26.56%), and discouragement of exploratory research (13.28%) were also mentioned as potential long-term negative consequences of mandatory preregistration (open text input item). In addition, researchers were concerned that others would not use preregistration correctly if they were obliged to do it (12.03%).

Lower motivation to preregister in participants with preregistration experience (open text input item) aligned with these arguments: Loss of flexibility (26.67% of 15 responses), time costs (20%), and that it was not suitable for their research (20%). Participants also stated that it was difficult to justify deviations to reviewers and editors (13.33%) and that their data was oftentimes unpredictable (13.33%). Reasons given by participants without preregistration experience for not using it (open text input item) were also largely consistent with the general

**Table 2. Comparison of worries and actual problems regarding preregistration.**

| "What worries do you have with respect to preregistering your studies?" | Percentage (Frequency) *N* = 94 | "Did you encounter specific problems when preregistering a study? If yes, which ones?" | Percentage (Frequency) *N* = 190 |
|---|---|---|---|
| High time costs | 61.7 (58) | It took very long to do the preregistration | 45.26 (86) |
| Low flexibility | 54.26 (51) | I found it problematic to not have flexibility during my analyses | 15.26 (29) |
| Maybe the study design would need to be changed because details do not work, but this would not be possible | 46.81 (44) | Study design would have needed to be changed because details did not work, but this was not possible | 13.68 (26) |
| If deviations were necessary, my study would lose credibility | 44.68 (42) | Deviations were necessary and my study lost credibility | 17.37 (33) |
| Scooping (i.e., someone taking my idea and publishing it before me) | 37.23 (35) | I got scooped (i.e., someone took my idea and published it before me) | 4.74 (9) |
| I would be insecure about what needs to be included in the preregistration | 29.79 (28) | I was insecure about what needs to be included in the preregistration | 41.05 (78) |
| Errors in the preregistration cannot be changed afterwards | 26.6 (25) | Errors in the preregistration could not be changed afterwards | 25.26 (48) |
| My supervisor/co-author(s) would object | 14.89 (14) | Conflict with supervisor/co-author | 9.47 (18) |
| Other | 5.32 (5) | Other | 12.11 (23) |
| None | 3.19 (3) | None | 13.68 (26) |

"Worries" item was shown to participants without preregistration experience, and "problems" item was shown to participants with preregistration experience. Multiple options could be selected.

drawbacks like time costs (30% of 40 responses) or the belief that no changes are possible after preregistering a study (17.5%). Additional points included a lack of knowledge (20%), their supervisor or co-workers being against it (17.5%), or overall costs (10%). Some people also wanted to wait and see how preregistration develops (5%).

**Comparison of worries and actual problems.**   In addition to examining these overall benefits and drawbacks, we also directly compared the worries of researchers who had not yet preregistered with the problems encountered by participants when preregistering their study in the past (multiple choice items, see Table 2). Worries associated with preregistering were, above all, high time costs (61.7% of 94 responses), low flexibility (54.26%), and the inability to deal with deviations, that is, that changes might not be possible (46.81%) or would lead to decreased credibility (44.68%). However, while the time costs were also the most indicated problem encountered by researchers that had preregistered (45.26% of 190 responses), low flexibility (15.26%) and deviations (changes not possible: 13.68%; loss of credibility: 17.37%) were not considered as problematic. Instead, researchers were more insecure what aspects needed to be included in the preregistration (41.05%). Additional problems experienced by the sample, as indicated by the open text input, comprised problems concerning changes after the preregistration (6.32%), problems with preregistration templates (2.11%), or a lack of understanding among reviewers (1.58%).

**Proportion of participants that read preregistrations.**   When asked how often they read the corresponding preregistration of a paper (single choice item), only about a third of participants indicated that they always (4.57% of 328 responses) or most of the time (24.09%) did so. The remaining participants either rarely (39.63%) or never (18.9%) read the corresponding preregistration to a paper, or even do not recall reading a paper that was preregistered (12.8%, see Fig 4). Specifically, while only few participants that had preregistered before could not recall reading a paper that was preregistered (5.02% of 219 responses), for participants without preregistration experience this was more common (28.44% of 109 responses).

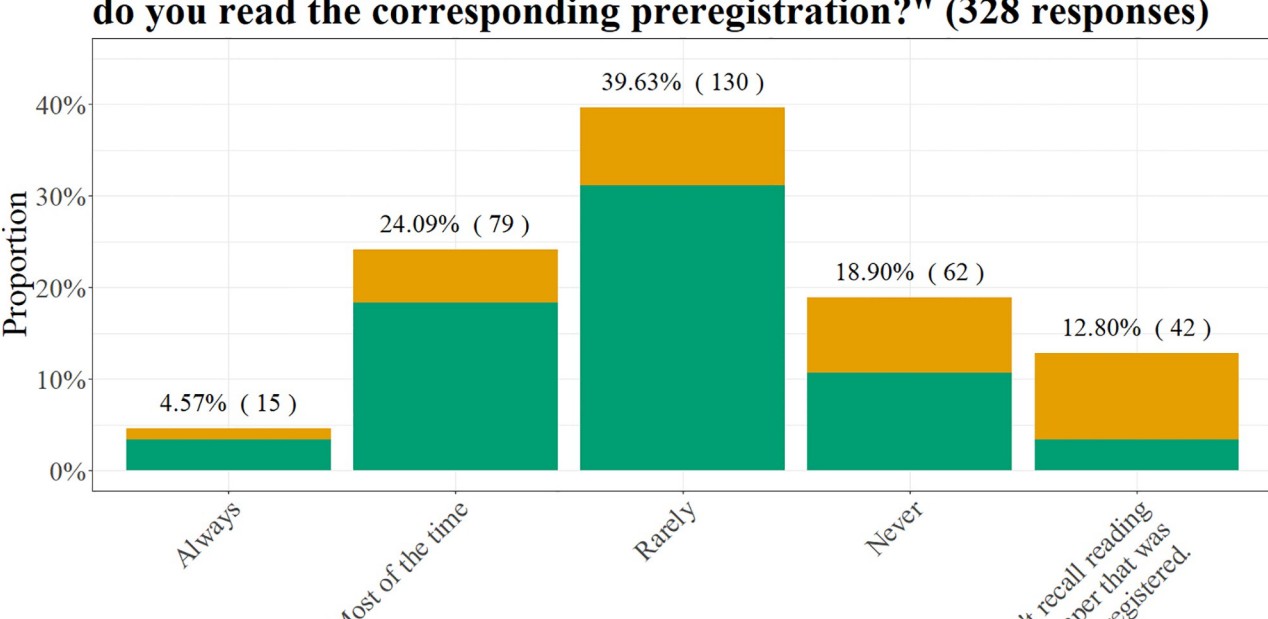

**Fig 4. How often do participants read preregistrations.** Above each bar, the percentage (frequency) of participants who selected the respective answer is shown.

**Persons and institutions that influence the decision to preregister.** Additionally, participants were asked which other persons or institutions influence their decision to use or not to use preregistration (multiple choice item). For early career researchers (i.e., participants with a bachelor's or master's degree), their supervisor was the biggest influence, while for the more senior groups, peers/colleagues and co-authors were considered the most influential. Overall responses as well as differences between academic groups and participants with and without preregistration experience are displayed in Fig 5. Other important influences, as indicated in the "other" comments, comprised researchers' own scientific and moral values (1.83%) or students (0.61%).

**Research experience and wish to stay in or leave academia.** We asked participants how long they had already worked in research and whether they planned to stay in or leave academia (single choice item) and compared these between participants with and without preregistration experience. On average, participants had already worked in research for about ten years, though there was much variability (*Mean* = 10.08, *Median* = 6, *SD* = 10.07, *IQR* = 12, *range* = 60; with preregistration experience: *Mean* = 10.4, *Median* = 7, *SD* = 9.87, *IQR* = 11, *range* = 60; without preregistration experience: *Mean* = 9.34, *Median* = 5, *SD* = 10.52, *IQR* = 10.5, *range* = 45). When participants with preregistration experience were asked about their career plans, 85.19% planned to continue their academic career, while 12.04% considered leaving academia (216 responses). For participants without preregistration experience, 75.45% planned to stay in academia and 16.36% considered leaving (110 responses). The remaining participants responded "other".

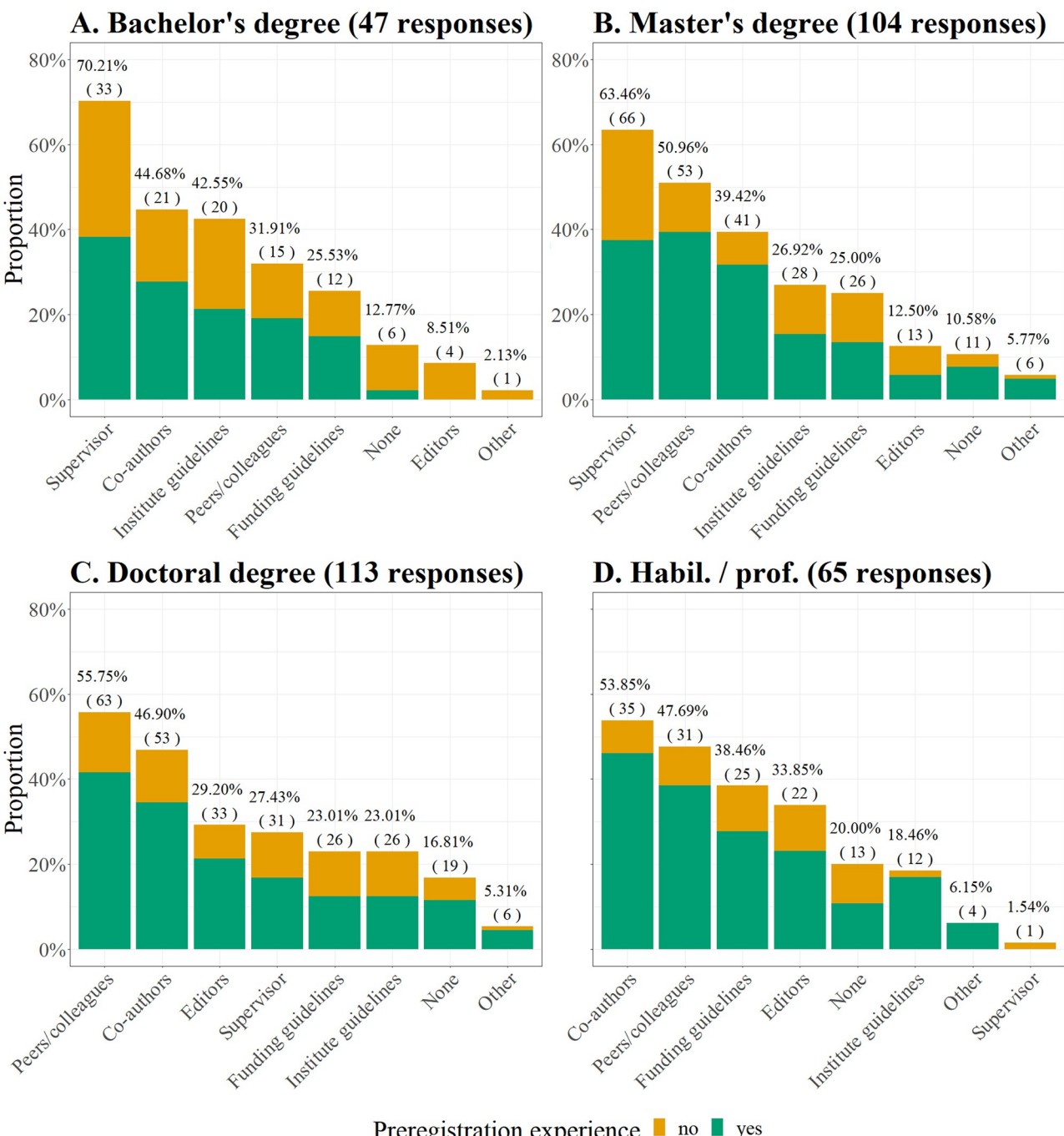

**Fig 5. Parties that influence participants' decision for or against preregistration.** Results are displayed separately for participants with different degrees (A-D). Above each bar, the percentage (frequency) of participants who selected the respective answer compared to all participants that responded to the item is shown. With preregistration experience: A) $n = 23$, B) $n = 70$, C) $n = 79$, D) $n = 48$. Without preregistration experience: A) $n = 24$, B) $n = 34$, C) $n = 34$, D) $n = 17$. Multiple options could be selected.

**Suggestions to improve preregistration.** In the final section of the survey, participants were asked for suggestions to increase the motivation and reduce obstacles of preregistration, and to improve the preregistration process in various open text input items.

The main ideas to increase the motivation to preregister were better incentives (e.g., funding, advantages for publication, job applications or tenure; 16.79% of 131 responses), providing more education (14.5%), and making preregistration mandatory (14.5%). To reduce obstacles to preregister, education was also the most frequently mentioned suggestion (27.78% of 108 responses). Other ideas included taking off pressure, for example, by destigmatizing deviations (12.04%), and providing clear and easily accessible templates (11.11%). An overview of all suggestions to increase the motivation and decrease obstacles to preregister is given in Fig 6.

Regarding suggestions to facilitate the preregistration process, participants were first asked about template improvements. Provided ideas included, above all, easy usability (18.18% of 55 responses), clear guidelines and standardization (16.36%), and good examples (12.73%). Furthermore, participants suggested a more automated process with many possibilities (e.g., including tables and figures; 10.91%). Specialized templates should be available, fitting different research situations (9.09%), and ideally with multiple levels of complexity (e.g., short templates where extra items can be added depending on the study, and "other" options; 5.45%) which allow for much flexibility (e.g., by also providing a "not yet planned" option; 5.45%). When asked directly whether they would prefer a more open or automated preregistration process (single choice item), participants expressed that they preferred a more restricted template that gives a lot of suggestions and reminds you of left-out information (53.19% of 188 responses) over a more open template where you are free to write what you want (34.57%). Furthermore, 6.38% of participants do not use templates at all and write their own text, 2.13% indicated that they preferred an open template with details or a mix of both answers, and 1.06% indicated that they had no preference or made this decision based on their expected data. Regarding the preregistration process (single choice item), participants also preferred a more automated, computer-assisted (57.84% of 185 responses) over a more open, self-administered process (37.3%). Again, some participants preferred a combination of both (1.62%) or said this depended on their project (1.08%).

Enhancing the usability was also often mentioned as a suggestion for improving preregistration repositories (47.62% of 42 responses), with user-friendliness and possible integration into a central network being the main points mentioned. Additionally, simple tracking of deviations (e.g., by enabling timestamped revisions) was emphasized (11.9%).

Regarding the review process, participants suggested making it as quick as possible (12.77% of 47 responses). Additionally, the preregistration should be checked against the final manuscript (10.64%), while deviations (10.64%) and null results (6.38%) should be more accepted. Lastly, participants highlighted the need for detailed and clear feedback during the reviewing and revisions process (6.38%).

To better integrate preregistrations into papers, participants suggested to make preregistration mandatory (23.08% of 39 responses), and to display the link to the preregistration more prominently in the paper (23.08%). Open science badges [75] that are displayed prominently in research articles and signal to readers which open science techniques are associated with this article (e.g., preregistration) were also mentioned (15.38%). These already exist today. In addition, participants requested that deviations should be clearly described (7.69%) using explicit guidelines (5.13%), and that in general it should be clearer what was preregistered and what was not (5.13%).

Lastly, ideas were requested for education on preregistration. Here, it was suggested as a priority that preregistration should be integrated into the university curriculum (36.54% of 52

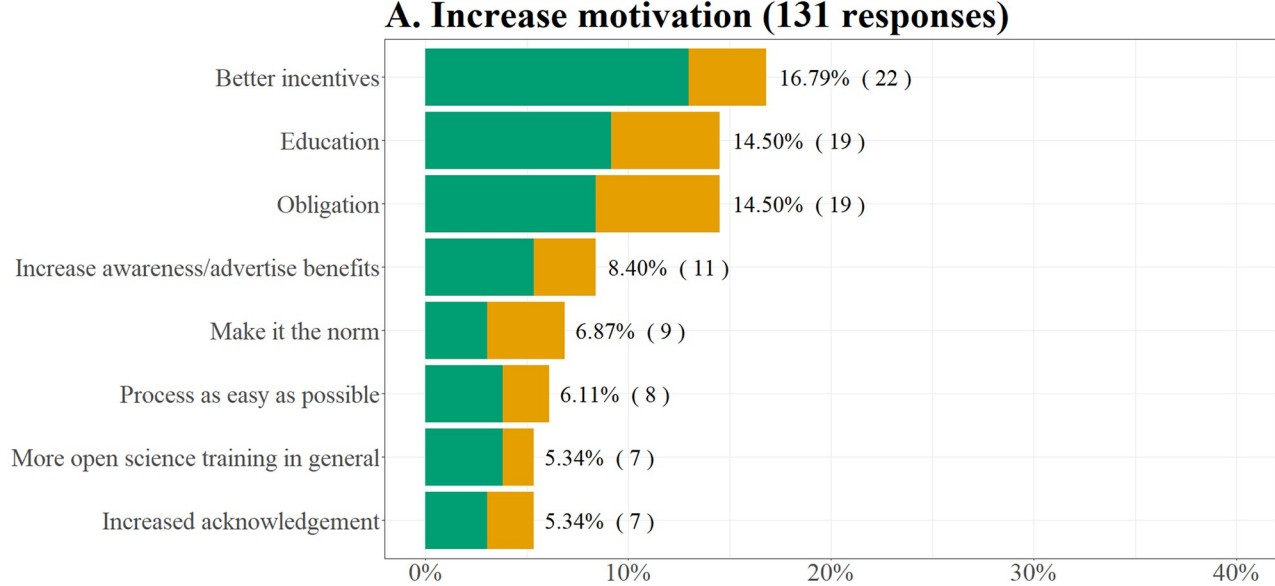

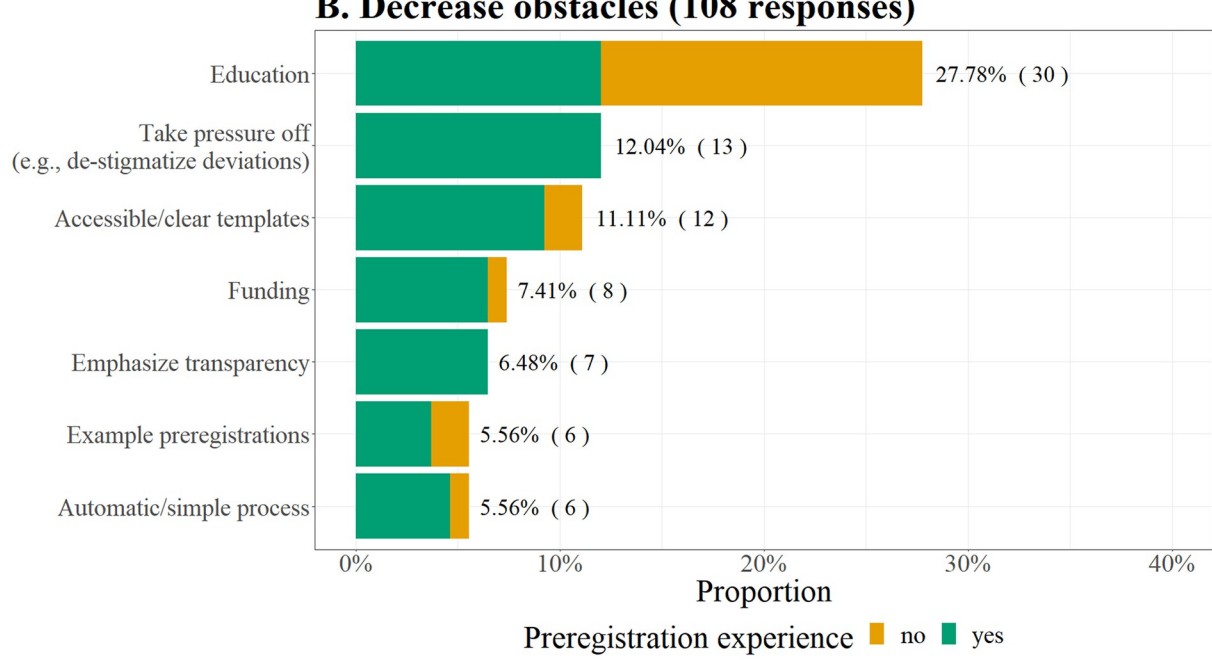

**Fig 6.** Suggestions to A) increase the motivation and B) decrease obstacles to preregister. Next to each bar, the percentage (frequency) of participants who indicated the respective theme is shown. A) With preregistration experience: *n* = 78, without preregistration experience: *n* = 53. B) With preregistration experience: *n* = 68; without preregistration experience: *n* = 40.

responses) and that there should be mandatory courses and workshops (34.62%). Improved accessibility of education resources, example preregistrations and clear templates, and guidance on how to deal with deviations were specific suggestions to improve researchers' knowledge about preregistration (each 5.77%).

Table 1 in S6 Text provides an overview of participants' suggestions for improving preregistration templates, repositories and publication, review, integration into papers, and education, displayed separately for participants with and without preregistration experience.

**Table 3. Overview of scales used for the hypotheses tests.**

| Scales | With preregistration experience $N = 194$ | Without preregistration experience $N = 95$ | Overall $N = 289$ |
|---|---|---|---|
| Attitude scale | 1.27 (0.92) | 0.8 (1) | 1.11 (0.97) |
| | *range* = 4.67 | *range* = 4.75 | *range* = 4.83 |
| Subjective norm scale | 0.78 (0.77) | 0.03 (0.83) | 0.54 (0.87) |
| | *range* = 3.75 | *range* = 4.25 | *range* = 4.38 |
| Perceived behavioral control scale | 1.49 (0.97) | 0.04 (0.91) | 1.01 (1.17) |
| | *range* = 4 | *range* = 4.2 | *range* = 4.8 |
| Intention scale | 1.94 (1.11) | 0.7 (1.47) | 1.53 (1.37) |
| | *range* = 5 | *range* = 6 | *range* = 6 |
| Motivation scale | 0.84 (0.87) | 0.42 (0.98) | 0.7 (0.93) |
| | *range* = 5.3 | *range* = 4.9 | *range* = 5.4 |
| Obstacle scale | -0.7 (1.03) | 0.11 (1.01) | -0.43 (1.09) |
| | *range* = 4.5 | *range* = 4.9 | *range* = 5.6 |

The following parameters are displayed: *Mean* (*SD*), *range*. Scales ranged from 1 = "Strongly disagree" to 7 = "Strongly agree" and were recoded to "-3 to +3" for data analyses. The parameters in this table were calculated based on the sample used for the hypotheses tests (i.e., complete datasets).

## Hypotheses tests

To find out how the intention to preregister is formed, and whether research experience has an impact on attitudes and the intensity of motivations and perceived obstacles regarding preregistration, we analyzed participants' responses to the attitude, subjective norm, perceived behavioral control, intention, motivation, and obstacle scale (see S3 Script for all analyses). Only complete datasets (see section *exclusion*, *missing data*, *and sample size*) were considered for these analyses. Before the analyses, the scales were inspected (see section *data analysis and preprocessing*), and the mean, standard deviation, and distribution of responses were displayed per item in tables for both the general and the database sample (i.e., only including participants recruited via the databases) for easy inspection (see S7 Text). Next, means, standard deviations, and ranges were computed for each scale (see Table 3).

A significance level of $\alpha$ = .05 was used for our hypotheses tests. Before conducting statistical analyses, assumptions were tested for each method (see S2 Script), all of which were fulfilled. If assumptions had been violated and tests had not been robust against these violations, alternative methods would have been used as described in the Registered Report Protocol (DOI: 10.1371/journal.pone.0253950). Furthermore, it was defined a priori how different result patterns would be interpreted, which was also described in the Registered Report Protocol.

For hypothesis 1, we predicted that attitudes, subjective norms, and perceived behavioral control influence researchers' intention to preregister their studies in the near future. Perceived importance of the topic of preregistration was included as a moderator for attitudes, and previous preregistration experience served as a control variable. To test this hypothesis, a moderated multiple regression model was computed: The intention (i.e., mean score of three items) was included as dependent variable. As predictors, the attitude scale (i.e., mean score of 24 items inquiring about how positive preregistrations are perceived), perceived importance of preregistration (one item), and their product term, the subjective norm scale (i.e., mean score of eight items inquiring about how beneficial participants perceive the social norm of using preregistration), the perceived behavioral control scale (i.e., mean score of five items inquiring about perceived controllability of the behavior), and previous preregistration experience (one

item: yes vs. no) were included. We expected that higher scores on the attitude scale (i.e., more positive attitudes, see *hypothesis 1.1*), higher scores on the subjective norm scale (i.e., higher perceived social pressure, see *hypothesis 1.4*), higher scores on the perceived behavioral control scale (i.e., higher perceived control, see *hypothesis 1.5*) are positive predictors for participants' intention scores (i.e., higher intention to preregister their studies in the near future). Additionally, we expected that the perceived importance of preregistration is a positive predictor for intention (see *hypothesis 1.3*), and significantly moderates the strength of the influence of attitudes positively (see *hypothesis 1.2*). Furthermore, we expected that the overall model including all predictors can significantly predict researchers' intention to preregister their studies in the future (see *hypothesis 1.6*). Because of these directional hypotheses, the regression weights were tested in a one-tailed fashion. Standardized regression weights were interpreted and compared. Preregistration experience was included into the model to control for a potential sampling bias. We had no strong predictions regarding the direction of this effect, since it might be the case that a) researchers who have used preregistration before are more likely to preregister again or b) that their preregistration experiences were negative, and they will less likely preregister again. Therefore, in contrast to the other predictors, this effect was tested in a two-tailed fashion.

As expected, researchers' attitudes (*hypothesis 1.1*), $t(282) = 10.07$, $p_{\text{one-sided}} < .001$, $\beta = .457$, subjective norms (*hypothesis 1.4*), $t(282) = 3.87$, $p_{\text{one-sided}} < .001$, $\beta = .148$, and perceived behavioral control (*hypothesis 1.5*), $t(282) = 4.65$, $p_{\text{one-sided}} < .001$, $\beta = .199$, significantly predicted their intention to use preregistration positively. Furthermore, perceived importance of preregistration was a significant positive predictor for intention (*hypothesis 1.3*), $t(282) = 7.16$, $p_{\text{one-sided}} < .001$, $\beta = .355$. Overall, attitudes and perceived importance were the strongest predictors of intention. In contrast to our hypothesis, perceived importance did not moderate the effect of attitudes on intention positively (*hypothesis 1.2*), $t(282) = -3.92$, $p_{\text{one-sided}} > .999$, $\beta = -.175$. Instead, the interaction effect was negative. When tested two-sided (as specified a priori), preregistration experience did not significantly predict the intention to use preregistration, $t(282) = 1.87$, $p_{\text{two-sided}} = .062$, $\beta = .071$. As expected, the overall model significantly predicted the intention to preregister in the future (*hypothesis 1.6*), $F(6, 282) = 137.6$, $p < .001$, explaining 73.99% of the dependent variable's variance (adjusted $R^2$).

The second hypothesis predicted that the research experience (i.e., the years someone has worked in research) influences researchers' attitudes (see *hypothesis 2.1*) as well as motivations (see *hypothesis 2.2*) and the perception of obstacles (see *hypothesis 2.3*). To investigate this, three multiple regression models were computed: Mean scores on the attitude scale (i.e., mean score of 24 items inquiring about how positive preregistrations are perceived), motivation scale (i.e., mean score of ten items measuring how strongly participants agree with potential motivations to preregister), and obstacle scale (i.e., mean score of ten items measuring how strongly participants agree with potential obstacles to preregister) served as dependent variables in the respective regression models, while research experience (i.e., the years someone has worked in research) and the preregistration experience (control variable) served as predictors. As our hypotheses regarding the impact of research experience on attitudes and the intensity of motivations and perceived obstacles were non-directional, the regression weights were analyzed with two-tailed tests. Since three regressions were conducted to test one set of hypotheses, a Bonferroni-Holm correction was used to adjust the alpha level.

As predicted, the research experience significantly influenced researchers' attitudes (*hypothesis 2.1*), $t(286) = -5.04$, $p_{\text{cor}} < .001$, $\beta = -.279$, as well as their perception of motivations to preregister (*hypothesis 2.2*), $t(286) = -5.03$, $p_{\text{cor}} < .001$, $\beta = -.279$. However, it had no effect on the perception of obstacles (*hypothesis 2.3*), $t(286) = 0.49$, $p_{\text{cor}} = .623$, $\beta = .027$. The influence of research experience on attitudes and motivations was negative, implying that

researchers who have only worked a small amount of time in research have more positive attitudes and feel more motivation to preregister than individuals who have worked in research longer. In contrast, obstacles were perceived the same regardless of the length of research work. Preregistration experience was a significant predictor for attitudes, $t(286) = 4.38$, $p_{cor} <$ .001, $\beta = .242$, and for the perception of motivations, $t(286) = 4.08$, $p_{cor} < .001$, $\beta = .226$, and obstacles, $t(286) = -6.26$, $p_{cor} < .001$, $\beta = -.348$. Participants with preregistration experience had more positive attitudes and perceived motivations to preregister more strongly, while they agreed less with the presented obstacles. Research experience and preregistration experience combined explained 12.29% of the variance of attitudes, 11.59% of the variance of motivations, and 11.45% of the variance of obstacles (adjusted $R^2$, all $p_{cor} < .001$). For a clear overview, the results of all regression models are presented in Tables 1 to 4 in S8 Text.

## Discussion

In the present study, we aimed to explore thoughts, motivations, and perceived obstacles of psychological researchers toward preregistration. Specifically, we wanted to shed light on the discrepancy between public support for preregistration in psychology on the one hand, and the concurrent low levels of preregistered studies on the other hand. In addition, we aimed to identify potential barriers to the use of preregistration which should be addressed to foster this open science technique.

### Positive perception of preregistration

In our sample, around 62% of researchers reported having preregistered before. This fits with the positive trend described by Christensen et al. [24], according to which in 2017, 44% of the psychological researchers they surveyed had preregistered at least once (however, it could also indicate a sampling bias, see section *limitations*). In our survey, almost 50% of those with a bachelor's degree had already preregistered, and in the more senior research groups the proportions were even higher. On average, preregistration was perceived positively, and most researchers planned to preregister their studies in the future. Specifically, 83.6% of participants had scores above zero on the intention scale, with zero indicating an undecided opinion and positive values indicating they were more likely to intend to preregister in the near future. It was even proposed by many participants that preregistration should be made mandatory.

The benefits of preregistration indicated by our sample were associated with improved planning and documentation, transparency, higher quality, and replicability of research, which is consistent with the arguments for preregistration found in the literature (e.g., see [15, 16, 29]). The overall positive outlook found in our survey aligns with recent studies by Logg et al. [27] and Sarafoglou et al. [76] which also showed mostly positive attitudes of researchers regarding preregistration when inspecting generational differences and the perceived impact of preregistration on the research workflow.

### Perceived obstacles and how they might be overcome

Various obstacles were identified that currently discourage psychological researchers from using preregistration, many of which align with arguments made by critics of preregistration [15, 16, 28, 30–32]. For some of these, solutions already exist. Others can be used as a starting point for improvement, as outlined below.

**Effort and time.** Perceived effort and time costs were the most voiced obstacles of preregistration in our survey, reported both as worries by individuals without preregistration experience (61.7%) and as actual problems encountered by researchers with preregistration experience (45.26%).

To address these obstacles, we suggest a three-folded approach: 1) Emphasize that and how the effort spent on the preregistration will be useful later on, 2) reduce the effort, or 3) reward it accordingly.

Thus, first, it should be better communicated to researchers that the time and work they put into their preregistration is not wasted. In theory, preregistration should not cause more effort but should rather shift the effort of writing the manuscript to an earlier point in time. Of course, one might argue that it imposes more work to consider all possibilities beforehand that might arise during the data analyses. However, if one does not plan in as much detail before conducting the study and instead makes the decisions ad hoc when analyzing the data, this may be faster, but decisions may easily be biased by the data, in which case the interpretations no longer have the same validity. Thus, preregistration may require more effort, but this is common for rigorous methods. The question is how to best utilize this effort and to maximize the outcomes, thus increasing the efficiency of preregistration. For example, it may be advantageous to increase the fit between preregistration and manuscript by aligning the information in the preregistration with established journal article reporting standards like JARS-Quant [77], which is the case for the PRP-QUANT Template (available at https://doi.org/10.23668/psycharchives.4584 [78]). This way, the preregistration template serves as checklist when planning a study and can be utilized as a scaffold when writing the final report of the results.

Second, better education and training could further help reduce the perceived effort of preregistration, because the more researchers know about how to prepare their preregistration, the less time it will likely take them (see section *lack of knowledge*).

And third, better incentives for preregistration should be provided to reward the efforts of preregistering researchers. This was also the most common suggestion from our participants to increase the motivation to preregister. They suggested, for example, to provide funding or advantages in publishing and job applications. That the incentive structure in academia needs to be revisited has long been a debated point (see, e.g., [22]). One example of such a reform is the publication format of "Registered Reports", where introduction and methods sections are submitted and peer-reviewed before the data collection of a study starts (i.e., a peer-reviewed preregistration). Registered Reports create a direct incentive through in-principle acceptance based on an evaluation of the research aim and methods [79], meaning that if accepted, the study will be published regardless of its results.

Another opportunity to reward the effort associated with preregistering would be for researchers to receive feedback from others on their preregistrations before collecting data, which is potentially more useful than only receiving it after the study was conducted. One example of a platform where preregistrations (and stage 2 protocols) are peer-reviewed is PCI-RR (https://rr.peercommunityin.org/). Here, experts are invited to review their peers' preregistrations and provide them with feedback that could be used to improve their studies. This open exchange can also facilitate collaborations, in that interested researchers could contact preregistration authors to collaborate in their project. At the Leibniz Institute for Psychology (ZPID), we offer the services PreReg and PsychLab that are disciplinary to psychology and more restricted in scope but combine the benefits of peer feedback and funding: Researchers can apply for a free-of-charge data collection with their preregistration (https://www.prereg-psych.org/index.php/rrp/lab-track). The preregistrations are peer-reviewed by external reviewers, and feedback can be incorporated before starting the study (similar to a Registered Report [79]). The platform is not linked to a journal, i.e., authors are free to select where they want to publish their study.

**Low flexibility and inadequate dealing with deviations.** Limited flexibility due to preregistration was another commonly reported obstacle, indicated by 17.58% of the sample as a drawback of this open science technique. Among participants without preregistration

experience, 54.26% worried about low flexibility and 46.81% were concerned that their study design might need to be changed after the preregistration which would not be possible.

Indeed, a certain degree of restriction of flexibility is part of the basic idea of preregistration (as argued, for example, by Wicherts et al. [80] who propose a checklist to assess the quality of preregistrations based on their ability to decrease researcher degrees of freedom). Much of the problem, however, is likely due to researchers' continued concern that they must adhere one hundred percent to the preregistration and that deviations from preregistered plans may result in a loss of credibility for their study. This worry was expressed by 44.68% of the researchers with no preregistration experience. Strikingly, this does not seem to be an unwarranted concern, as 17.37% of the participants with preregistration experience reported having encountered this problem. Consequently, disclosure of deviations from preregistered plans in finished manuscripts is currently inadequate [34–37]. This insufficient handling of deviations and fear of authors to disclose deviations in articles is especially problematic because, as indicated in our survey, researchers do not typically read the corresponding preregistration to an article. Thus, the preregistration badge or statement can create a false sense of confidence in a study's results if people think that everything has been done as preregistered, when in fact deviations were simply not disclosed.

Thus, destigmatizing deviations could encourage more people to preregister, reduce much uncertainty about this technique, and ensure that it has the intended positive effect on transparency. Correspondingly, taking pressure off, for example, by destigmatizing deviations, was also the second most common suggestion from participants for reducing obstacles of preregistration. This objective could be achieved by emphasizing that deviations are normal (e.g., see "Preregistration: A Plan, Not a Prison" [81]) and by establishing a standardized way of reporting them (e.g., in the form of a supplementary material).

**Lack of knowledge.** Next, uncertainty about what to include in the preregistration was the second most common problem of researchers who had already preregistered (reported by 41.05%), and a worry of 29.79% of researchers without preregistration experience.

Correspondingly, better education was the second most frequent suggestion for increasing the motivation to preregister, and the most frequent suggestion for reducing obstacles of preregistration. Specifically, participants suggested incorporating preregistration into the study curriculum. For example, this could be done by including preregistration in the practical research internships at university, where students conduct studies themselves (e.g., see [82]). Other provided ideas included mandatory courses and workshops and improved accessibility of teaching materials.

Additionally, participants highlighted providing accessible and clear templates. For this, a recent effort of various large psychological societies (i.e., the American Psychological Association, the British Psychological Society, and the German Psychological Society) in cooperation with the Center for Open Science and the Leibniz Institute for Psychology (ZPID) represents an already existing solution. Together, they created a comprehensive template, the Psychological Research Preregistration-Quantitative (PRP-QUANT) Template (available at https://doi. org/10.23668/psycharchives.4584 [78]), which is modeled on a typical research article and includes detailed instructions for filling in the information. In addition to this comprehensive and universal template, researchers should also be made aware that there are a variety of templates for specific research branches (e.g., for fMRI studies [83] or secondary data analyses [84]). Learning about these specialized templates could also reduce researchers' worry that preregistration is not suitable for all projects. Overall, appropriate templates could support researchers in filling knowledge gaps and further reduce uncertainty.

**Exploratory research.** Another perceived obstacle of preregistration is the devaluation and discouragement of exploratory research, reported by 7.14% our participants as a drawback

of preregistration, and by 13.28% as a possible negative consequence of mandatory preregistration.

Indeed, preregistration mainly addresses confirmatory studies and aims to distinguish confirmatory from exploratory research. However, it is possible to preregister exploratory research questions (e.g., in the PRP-QUANT Template [78]), which might be advantageous, as argued, for example, by Dirnagl [85] in the context of biological research. Nevertheless, researchers may remain concerned that preregistration favors confirmatory studies and disadvantages exploratory research. For this reason, a new article category called "Exploratory Reports" has been introduced in an increasing number of journals (e.g., in Cortex [86]) as a complement to Registered Reports. By explicitly focusing on exploratory studies, these aim to promote exploratory research.

**Scooping.** Lastly, over a third of participants with no preregistration experience indicated being afraid of scooping (37.23%). Of the researchers who had already preregistered, 4.74% reported having been scooped. As part of our exploratory analyses, we took a closer look at the nine scooped participants (see S5 Text). Overall, their attitudes seemed less positive, they perceived motivations somewhat lower, and obstacles somewhat higher than researchers who were not scooped. Likewise, intention scores were slightly lower on average (however, no significance tests were performed because the group of scooped researchers was so small). On a positive note, however, six of the nine scooped researchers still tended to want to preregister again, that is, they had scores above zero on the intention scale, which ranged from -3 to 3 (recoded) and where zero represented an undecided opinion.

Researchers concerned about being scooped may be advised to put an embargo on their preregistration as a solution to this problem. In this case, the preregistration is still recorded online with a timestamp, but is only visible to others after a certain date, which can be determined by the preregistering researcher. In this way, it is still possible to transparently show in retrospect at which time point certain decisions have been made, without the study plan being directly accessible by others. An embargo period can be set up on all common preregistration platforms in psychology.

## Facilitating factors and influence of research experience

Besides these descriptive insights into the psychological research community, we investigated two specific research questions. First, we wanted to identify which factors facilitate or prevent the usage of preregistration. We expected that the predictors of the theory of planned behavior (attitudes, subjective norm, and perceived behavioral control) [40, 41] as well as the perceived importance of preregistration (and its interaction with attitudes) influence researchers' intention to preregister. Our results show that, except for the expected interaction effect, all predictors indeed significantly influenced researchers' intention to preregister their studies, with a huge proportion of variance of intention being explained ($R^2$ = 73.99%). Thus, we conclude that the theory of planned behavior is indeed well suited to be used to predict researchers' intention to preregister, with attitudes and perceived importance being the strongest predictors. Therefore, to promote preregistration, it might be especially beneficial to highlight the relevance of preregistration for each individual researcher, for example, by emphasizing individual benefits like the usefulness for planning, which was the highest ranked benefit reported by participants in our survey.

Second, we examined whether research experience, i.e., the time someone has worked in research, influences attitudes and the perceived intensity of motivations and obstacles. Indeed, research experience was a significant predictor of both attitudes and the perceived intensity of motivations, while it did not significantly influence the perception of obstacles. More

specifically, it was the case that psychologists newer to research had more positive attitudes toward preregistration and agreed more with statements expressing a high motivation, whereas obstacles were perceived independently of the research experience. These results could be interpreted as supporting the assumption that early career researchers are more invested in preregistration than more senior researchers. However, in our sample, 71.67% of participants with a habilitation or full professorship had already preregistered and for the most part also valued preregistration as important (especially those who had already preregistered, see section *perceived importance and intention to preregister*). Therefore, it can be argued that attitudes toward preregistration tended to be positive for this group as well, only more so for early career researchers.

## Limitations

The main limitation of our study is a potential selection bias of individuals who are more positive towards preregistration or have preregistered before. We tried to counteract this by recruiting researchers who were identified through their articles in large databases. We also reported, wherever possible, the results of individuals with and without preregistration experience; excluded participants recruited through the OSF from reports intended to represent the overall psychology community; and included preregistration experience as a control variable in our analyses. Nevertheless, bias cannot be conclusively ruled out, and the proportion of participants with preregistration experience was very high in our sample (61.81%).

On a more general note, it should be borne in mind that that preregistration has a variety of benefits that help increase scientific rigor, but it is not a panacea (as argued by [87]). Therefore, preregistration should ideally be used together with complementary open science practices increasing transparency and confidence (see [87] for an overview). Even so, for comprehensive and long-term improvement of the credibility of scientific evidence, the academic system needs to be revised, for example, by incentivizing methodological quality instead of results.

## Future research

The theoretical benefits of preregistration are obvious and have been argued for by numerous authors (e.g., [see 15, 16, 21, 23]). However, these should be tested more thoroughly empirically to ensure that preregistration is used and works in the intended way and to realize its full potential. An example for this is the current inadequate reporting of deviations [34–37], which might compromise the positive effect of preregistration on transparency. Our survey provided further support that researchers worry about negative consequences after the occurrence and reporting of deviations. We suggest developing a standardized reporting method for deviations that should then be empirically tested to see if it leads to deviations becoming more transparently disclosed.

## Conclusion

Our survey allowed us to get a sense of what attitudes, motivations, and obstacles psychological researchers currently have toward preregistration. Various practical solutions already exist for many of the identified obstacles. Meanwhile, other problems, that were reported by both researchers with and without preregistration experience, should be further addressed and can serve as a basis for improvement. We showed that attitudes, subjective norm, perceived behavioral control, and perceived importance significantly influence the intention to preregister. Early career researchers had more positive attitudes and indicated higher motivation, whereas obstacles were perceived independently of research experience.

## Supporting information

**S1 Table. Survey items.** The items of the survey are presented. If they were derived from other studies, the original items as well as references are given.
(XLSX)

**S1 Video. Screen recording of the survey procedure (version A).** This screen recording shows the procedure of the survey for participants that responded with "yes" on the item "Have you preregistered a study before?" (as this is a filter question for some of the following items) and who answered the knowledge check of the preregistration definition correctly.
(MP4)

**S2 Video. Screen recording of the survey procedure (version B).** This screen recording shows the procedure of the survey for participants that responded with "no" on the item "Have you preregistered a study before?" (as this is a filter question for some of the following items) and who were shown the definition of preregistration again due to errors in the knowledge check.
(MP4)

**S1 File. Questionnaire.** The questionnaire is displayed in a PDF for easy inspection. Please note that filters do not work in this view, i.e., all items are displayed.
(PDF)

**S1 Text. Plain language summary.** A plain language summary is provided to increase the accessibility of these research findings to a wider audience. It is available in English and German.
(DOCX)

**S2 Text. Power analyses.** Power analyses for both hypotheses are presented here. For hypothesis 1, power analyses were performed for the entire model as well as for the individual predictors.
(DOCX)

**S3 Text. Data collection procedure.** The data collection procedure that is displayed in Fig 2 is further specified. Particularly, the specific procedures for collecting contact addresses as well as for inviting participants are described.
(DOCX)

**S4 Text. Research topics.** Research topics indicated by the sample are displayed in a table.
(DOCX)

**S5 Text. Exploratory analyses.** All exploratory analyses and results are described in this document.
(DOCX)

**S6 Text. Suggestions for improvements regarding the preregistration process.** Participants' suggestions for improving the preregistration process are displayed in a table.
(DOCX)

**S7 Text. Overview of survey items.** Tables containing information about the individual survey items (means, standard deviations, and distribution of responses) are provided here for both the general and the database sample (i.e., only including participants recruited via the databases).
(DOCX)

**S8 Text. Regression tables.** The results of all regression models used for the hypotheses tests are displayed in table format for easy inspection.
(DOCX)

**S1 Script. Preprocessing script.** This R script contains all steps of preprocessing. The hereby processed data were used for the assumption tests as well as for the descriptive reports and hypotheses tests.
(R)

**S2 Script. Assumption testing script.** This R script was used for testing the assumptions of the statistical models used to test hypotheses 1 and 2.
(R)

**S3 Script. Analysis script.** This script contains the code used to conduct all descriptive reports as well as the hypotheses tests of the survey.
(R)

**S4 Script. Exploratory analyses script.** This script includes the code for all exploratory analyses that were conducted to further inspect the data.
(R)

## Acknowledgments

We are grateful to H. Bargon, A. Fritz, A. Kuznik, D. Maurer, and M. Rauer for their help with testing the survey, recruiting participants, and coding, and to T. Dauber and J. Pauquet for testing the survey and proofreading.

## Author Contributions

**Conceptualization:** Lisa Spitzer.

**Data curation:** Lisa Spitzer.

**Formal analysis:** Lisa Spitzer.

**Investigation:** Lisa Spitzer.

**Methodology:** Lisa Spitzer.

**Software:** Lisa Spitzer.

**Visualization:** Lisa Spitzer.

**Writing – original draft:** Lisa Spitzer.

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
