## [Decision Letter · Decision Letter 0]

1 Dec 2022

PONE-D-22-30808Registered Report: Survey on attitudes and experiences regarding preregistration in psychological researchPLOS ONE

Dear Dr. Spitzer,

Thank you for submitting your manuscript to PLOS ONE. After careful consideration, we feel that it has merit but does not fully meet PLOS ONE’s publication criteria as it currently stands. Therefore, we invite you to submit a revised version of the manuscript that addresses the points raised during the review process.

Thank you for submitting this excellent stage 2 registered report. **I would like to thank the 3 reviewers who sent me a very fast feedback.** As you will see, all were very supportive of acceptance. There are however a few minor points that may help the manuscript for being even better. Please also note that one reviewer has attached a pdf of the manuscript with many annotations (in case you don't receive it, please let me know).

**I will be pleased to accept the manuscript after you carefully consider all these important suggestions (although those are minor and not mandatory changes). **

We look forward to receiving your revised manuscript.

Kind regards,

Florian Naudet, M.D., M.P.H., Ph.D.

Academic Editor

PLOS ONE

Journal Requirements:

Reviewers' comments:

Reviewer's Responses to Questions

**Comments to the Author**

1. Does the manuscript adhere to the experimental procedures and analyses described in the Registered Report Protocol?

If the manuscript reports any deviations from the planned experimental procedures and analyses, those must be reasonable and adequately justified.

Reviewer #1: Yes

Reviewer #2: Yes

Reviewer #3: Yes

2. If the manuscript reports exploratory analyses or experimental procedures not outlined in the original Registered Report Protocol, are these reasonable, justified and methodologically sound?

A Registered Report may include valid exploratory analyses not previously outlined in the Registered Report Protocol, as long as they are described as such.

Reviewer #1: Yes

Reviewer #2: Yes

Reviewer #3: Yes

3. Are the conclusions supported by the data and do they address the research question presented in the Registered Report Protocol?

The manuscript must describe a technically sound piece of scientific research with data that supports the conclusions. The conclusions must be drawn appropriately based on the research question(s) outlined in the Registered Report Protocol and on the data presented.

Reviewer #1: Yes

Reviewer #2: Yes

Reviewer #3: Yes

4. Have the authors made all data underlying the findings in their manuscript fully available?

Reviewer #1: Yes

Reviewer #2: Yes

Reviewer #3: Yes

5. Is the manuscript presented in an intelligible fashion and written in standard English?

Reviewer #1: Yes

Reviewer #2: Yes

Reviewer #3: Yes

6. Review Comments to the Author

Please use the space provided to explain your answers to the questions above. (Please upload your review as an attachment if it exceeds 20,000 characters)

Reviewer #1: Important note: This review pertains only to ‘statistical aspects’ of the study and so ‘clinical aspects’ [like medical importance, relevance of the study, ‘clinical significance and implication(s)’ of the whole study, etc.] are to be evaluated [should be assessed] separately/independently. Further please note that any ‘statistical review’ is generally done under the assumption that (such) study specific methodological [as well as execution] issues are perfectly taken care of by the investigator(s). This review is not an exception to that and so does not cover clinical aspects {however, seldom comments are made only if those issues are intimately / scientifically related & intermingle with ‘statistical aspects’ of the study}. Agreed that ‘statistical methods’ are used as just tools here, however, they are vital part of methodology [and so should be given due importance]. I look at the manuscript in/with statistical view point, other reviewer(s) look(s) at it with different angle so that in totality the review is very comprehensive. However, there should be efforts from authors side to improve (may be by taking clues from reviewer’s comments). Therefore, please do not limit the revision only (with respect) to comments made here.

COMMENTS: This manuscript of Registered Report of Survey is excellent, however, I wonder to note that the protocol with same name [Registered Report Protocol (DOI: 10.1371/journal.pone.0253950).] was published in July, 2021 but is mentioned directly in the ‘Methods’ section (i.e., in lines 191-92). There you said “methods and analyses reported were implemented as described in the protocol”. I find many similarities in these {this article and the protocol}. Very good that authors report them [good summary] but they are reported in supplementary file [S2 Text. Deviations from the Registered Report Protocol. This table summarizes all deviations from the preregistered procedure described in the Registered Report Protocol. For all, a description and justification are given], whereas, in my opinion, they should have been in main text.

As said in S2 “The section sampling procedure was moved down in the manuscript and divided into smaller subsections to facilitate readability of the article. In its current form, everything that was taken directly from the descriptions of the Registered Report Protocol is displayed in the beginning of the manuscript, and everything that was added (results, information about the data collection, e.g., proportion of participants recruited through each database etc.) is displayed in the second part of the manuscript” is useful. And what is said in lines 228-29 (also 534) [“Scales were recoded from “1 to 7” to “-3 to +3” for data analyses yielding a middle category which has absolute meaning”] is appreciated because that will definitely yield correct and meaningful ‘arithmetic mean’ which is useful not only for comparison but application of any statistical test(s) assumes that meaning of entity used (mean, SD, etc) has a particular meaning {which is achieved only then}.

As pointed out in ‘important note’ above “This review pertains only to ‘statistical aspects’ of the study and so ‘clinical aspects’ should be assessed separately/independently [one should carefully consider/look at the clinical implications of the study]. However, in my considered opinion, there should be no hesitation in accepting this article.

Reviewer #2: This Registered Report appears to be conducted and reported very thoroughly. I have read through the manuscript in its entirety and sifted through some of the supplementary materials. I have not read through the code or data, nor tried to rerun any of their analyses.

The manuscript would be okay to publish as is. Nonetheless, I do have comments that the authors may be interested in addressing. I have also attached a pdf of the manuscript with several more minor comment boxes I added while reading the manuscript.

1. The manuscript is long and detailed (which is good). However, this makes it difficult to extract the main messages of the manuscript. Additional figures/tables, or a “Results summary” section could be useful for the reader.

2. Figures 5 and 6 cannot be fully interpreted without additional information. For example, in Fig 6A, 22 of 131 people responded “Better incentives”. Some of these responses were from people who have preregistration experience and some were from people who have no preregistration experience. However, the percentage within each group is ambiguous. For example, it could be that 6/6 with no prereg experience selected “Better incentives” and 16/125 with prereg experience selected this. Alternatively, it could be 6/125 with no prereg experience and 16/16 with prereg experience. Perhaps including the total number with and without prereg experience in the footnote for each sub-figure would solve this issue.

3. There are many instances where the writing is ambiguous or unclear, particularly in the introduction. I have highlighted many of these sections in the attached pdf of the manuscript.

4. I find the table in S15 important. Perhaps the authors could consider moving it to the main manuscript. In particular, I feel another version of this table that presents only the results from the more random sample would be interesting (e.g., respondents identified by emails taken from databases – and not those from the OSF, mailing lists, or social media). This subset of the data could be compared to random samples in future research to see if things have changed. However, this comparison cannot be reasonably performed with the full sample, as this sample doesn’t represent a clear population (due to the sampling bias towards those recruited from the OSF, mailing lists, and social media who have a different attitude than a random sample—as evidenced in Table 1 of S13).

5. In the descriptive results section it is unclear which questions were open-text response, multiple choice with one selection option, and multiple choice with multiple selections possible. I recommend clarifying this throughout.

6. The denominator for how many people responded to a question changes for each question, and in some cases is very small compared to the total sample (e.g., 32). As a reader, it’s not clear why some questions have so few responses. Could you clarify this?

The PLOS manuscript portal asks me: “Does the manuscript adhere to the experimental procedures and analyses described in the Registered Report Protocol?” As far as I can tell, yes. However, to systematically check this would require many hours of (non-thrilling) work, which goes beyond my position as a volunteer peer reviewer. I ran a study specifically on peer reviewing registrations to manuscripts—it is labour intensive and often ambiguous.

Congrats on the thorough paper, it must have been a large effort. I hope it gets the traction it deserves.

I always sign my reviews,

Robert Thibault

Reviewer #3: The reviewer wants to congratulate the authors on this outstanding article.

Having been a reviewer for the protocol/ stage 1, the hard work and rigor put into this manuscript is much appreciated.

This article is close to being accepted but one comment from my side:

In the discussion section I am missing a point on how pre-registration is not to be treated as the "panacea" (as described by Hardwicke and Wagenmarkers in their pre-print https://osf.io/preprints/metaarxiv/d7bcu/ ). The limitation part of this article could benefit from discussing this point.

7. PLOS authors have the option to publish the peer review history of their article (what does this mean?). If published, this will include your full peer review and any attached files.

Reviewer #1: **Yes: **Dr. Sanjeev Sarmukaddam

Reviewer #2: **Yes: **Robert T. Thibault

Reviewer #3: No

---

## [Author Response · Author response to Decision Letter 0]

13 Jan 2023

Response to reviewers

We thank the reviewers for reading our revised manuscript thoroughly and for providing constructive comments and criticism. We believe that by addressing all points to our best knowledge, we were able to improve the manuscript further.

Following the reviewers’ comments and suggestions, we clarified remaining ambiguities in the text. Additionally, we now present the deviations from stage 1 more prominently as a table in the manuscript instead of in the supporting information. We revisited our discussion section and reference list, and we were able to correct some small errors and typos.

In addition to addressing the reviewer comments, we made the following changes: We updated the current R version, we corrected “p = 1” to “p > .999”, and we added to the descriptive results section one sentence reporting an option that was previously inadvertently omitted.

More details about the changes are included in our responses to each reviewers’ comments below. If line numbers were given by the reviewers, we include the term, sentence, or section, to which the comment refers to, in parentheses.

Reviewer #1

R: This manuscript of Registered Report of Survey is excellent, however, I wonder to note that the protocol with same name [Registered Report Protocol (DOI: 10.1371/journal.pone.0253950).] was published in July, 2021 but is mentioned directly in the ‘Methods’ section (i.e., in lines 191-92). There you said “methods and analyses reported were implemented as described in the protocol”. I find many similarities in these {this article and the protocol}. Very good that authors report them [good summary] but they are reported in supplementary file [S2 Text. Deviations from the Registered Report Protocol. This table summarizes all deviations from the preregistered procedure described in the Registered Report Protocol. For all, a description and justification are given], whereas, in my opinion, they should have been in main text.

Thank you for your comment. We have included the deviations overview table into the manuscript instead of providing it as part of the supporting information. It is now Table 1 in the manuscript.

R: As said in S2 “The section sampling procedure was moved down in the manuscript and divided into smaller subsections to facilitate readability of the article. In its current form, everything that was taken directly from the descriptions of the Registered Report Protocol is displayed in the beginning of the manuscript, and everything that was added (results, information about the data collection, e.g., proportion of participants recruited through each database etc.) is displayed in the second part of the manuscript” is useful. And what is said in lines 228-29 (also 534) [“Scales were recoded from “1 to 7” to “-3 to +3” for data analyses yielding a middle category which has absolute meaning”] is appreciated because that will definitely yield correct and meaningful ‘arithmetic mean’ which is useful not only for comparison but application of any statistical test(s) assumes that meaning of entity used (mean, SD, etc) has a particular meaning {which is achieved only then}.

As pointed out in ‘important note’ above “This review pertains only to ‘statistical aspects’ of the study and so ‘clinical aspects’ should be assessed separately/independently [one should carefully consider/look at the clinical implications of the study]. However, in my considered opinion, there should be no hesitation in accepting this article.

Thank you for your helpful feedback.

Reviewer #2

R: The manuscript is long and detailed (which is good). However, this makes it difficult to extract the main messages of the manuscript. Additional figures/tables, or a “Results summary” section could be useful for the reader. 

Thank you for your feedback. We agree that the manuscript is indeed very long, as we want to provide a comprehensive insight into psychological researchers’ perspective on preregistration. We have highlighted some of the most relevant findings in the discussion, for example, we were keen to discuss the identified obstacles. In addition, we have tried to make the rest of the results sufficiently clear by providing detailed sub-headings in the results section. Therefore, in order not to increase the length of the manuscript even further, we would like to refrain from adding an additional section or additional figures.

R: Figures 5 and 6 cannot be fully interpreted without additional information. For example, in Fig 6A, 22 of 131 people responded “Better incentives”. Some of these responses were from people who have preregistration experience and some were from people who have no preregistration experience. However, the percentage within each group is ambiguous. For example, it could be that 6/6 with no prereg experience selected “Better incentives” and 16/125 with prereg experience selected this. Alternatively, it could be 6/125 with no prereg experience and 16/16 with prereg experience. Perhaps including the total number with and without prereg experience in the footnote for each sub-figure would solve this issue.

Thank you for pointing out this issue. We have added the total number of participants with/without preregistration experience in the footnotes of these figures. While working on this comment, we were also able to correct a small error in the code for calculating the percentages of some items. Correcting this error changed a few of the reported percentages slightly, which were corrected everywhere. In addition, we discovered that we had not yet excluded inappropriate responses in one table (decrease obstacles), which we also corrected. We are glad that we were able to catch these errors thanks to your comment.

R: There are many instances where the writing is ambiguous or unclear, particularly in the introduction. I have highlighted many of these sections in the attached pdf of the manuscript.

Thank you for your detailed feedback, which was very helpful to us. We have addressed all the points raised in the PDF file and respond to them below.

R: I’d recommend the active voice to improve the clarity of the writing.

[L19: In a mixed-methods approach, an online survey was conducted, assessing attitudes, motivations, and perceived obstacles with respect to preregistration.]

We edited the sentence accordingly: “In a mixed-methods approach, we conducted an online survey assessing attitudes, motivations, and perceived obstacles with respect to preregistration.”.

R: through their email as corresponding authors?

[L21: Participants were psychological researchers that were recruited through their publications....]

We did not only contact the corresponding authors, but all authors listed in the publications/preregistrations. For this, we identified their contact information in an online search, as is described in the section “recruitment through different databases, email lists, and social media”.

R: This is several hypotheses...Can you break them down (or explain that you do later; or identify this as an overarching hypothesis)?

[L22ff: Based on the theory of planned behavior, we predicted that positive attitudes (moderated by the perceived importance of preregistration) as well as a favorable subjective norm and higher perceived behavioral control positively influence researchers’ intention to preregister (hypothesis 1).]

Because of the fit to the first part of the Registered Report, we would like to refrain from changing the description of the hypotheses in the abstract too much.

R: to reduce ambiguity this number should be placed after the word “participants”. And perhaps the term “respondents” should be used in place of “participants”

[L22: (N = 289)]

The sentence was changed accordingly.

R: was this hypothesis directional? clarify.

[L26f: Furthermore, we expected an influence of research experience on attitudes and perceived motivations and obstacles regarding preregistration (hypothesis 2).]

For both hypotheses 1 and 2, we added this information in the parentheses.

R: there were multiple elements to hypothesis 1. Are they all supported?

[L29ff: Researchers’ attitudes, subjective norms, perceived behavioral control, and the perceived importance of preregistration significantly predicted researchers’ intention to use preregistration in the future, thus supporting hypothesis 1.]

In the text, we name the sub-hypotheses that were supported. To make it clearer that not all of them were supported, we wrote “(see hypothesis 1)” instead of “thus supporting hypothesis 1”.

R: I don’t know if that’s the precise definition of replicability. consider rewording.

[L44f : replicability refers to the attempt to repeat an experiment to re-test the original effect]

To increase the clarity of this definition, we have chosen to use a different reference. In the current version, the definition is taken from the Open Science Glossary (Parsons et al., 2022), which we hope is more precise than the definition used before. The definition now reads: “where replicability refers to a study arriving at the same conclusion after collecting new data”.

R: I think it’s worth mentioning that this study had serious sampling bias.

[L45ff: conducted a survey of more than 1500 researchers of multiple disciplines, 70% of researchers reported that they had failed to replicate studies by others, and more than 50% had failed to replicate their own studies.]

We agree that sampling bias may have occurred in this survey (which is also directly addressed in the cited article). However, in our opinion, the quoted statements still adequately describe the fact that a large number of researchers perceived the problem of lack of reproducibility as early as 2016, even if a sampling error was present. Nevertheless, we added the term “surveyed” (“Overall, 90% of surveyed researchers indicated their belief in a slight or even significant crisis.”) to make it a bit clearer that this statement refers to the study’s respondents, not the general researcher population.

R: what do you mean by “failed”. There are multiple ways of interpreting this.

What’s the definitions they are using for “successful replication”.

[L50: Strikingly many attempts failed as shown by replication rates between 36 [2] and 77% [6].]

We have addressed these two comments by adding the bracket “failed (i.e., they did not yield significant effects)”. While inspecting the quoted publications, we noticed that the replication rate of 77% from source [6] we cited in our manuscript has some room for interpretation. In the cited article, the following is reported in the results section: “In the aggregate, 10 of the 13 studies replicated the original results with varying distance from the original effect size. One study, imagined contact, showed a significant effect in the expected direction in just 4 of the 36 samples (and once in the wrong direction), but the confidence intervals for the aggregate effect size suggest that it is slightly different than zero.” (p. 147-149). Based on this statement, the replication rate we reported was chosen (10/13 = 77%). However, in the discussion section of the original article, it is stated that 11 of the 13 effects could be replicated - so the slightly significant finding was also counted as successful replication here, which we originally pulled out differently. Based on this insight, we have adjusted the percentage in our manuscript and apologize for the misstatement.

R: or at least to make them detectable.

[L62: counter these questionable research practices]

We have added “to detect”: “The preregistration of studies has been proposed to detect and counter these questionable research practices”.

R: not necessarily. You can preregister an analysis for an existing dataset.

[L64: A preregistration is a research plan that is time-stamped, created before the data has been collected or examined]

This possibility was already addressed in this sentence by including “or examined”.

R: unclear who is saying it “needs” to be added.

[L67: If the research plan changes afterwards, either a new version needs to be added]

In our view, trustworthy preregistration platforms are responsible for requiring these new versions.

R: not necessarily. They are often quite difficult to detect. See van den Akker 2022.

[L67: or the deviations will be apparent when comparing the preregistration to the final manuscript]

We address the issue of poor reporting of deviations in more detail later in the manuscript. 

R: few preregistrations include this.

[L69f: Thus, preregistration aims for a transparent presentation of what was planned at a certain time point and what changes may have been made to a study until its publication]

Here, we are not referring to the reporting of deviations in the paper (although this is very relevant, as we discuss later in the manuscript), but to the general opportunity that preregistration provides to track changes through inspecting the preregistered procedure.

R: perhaps worth pointing out that these refs are to clinical studies, not psychology stdies.

[L71f: reduces questionable research practices and the rate of false positive findings (e.g., see [17,18]]

We have added “see ... for examples from clinical trials and epidemiology”.

R: define

[L74: Correspondingly, a recent study showed that preregistration, among other open science techniques, can drastically increase the replication rate [19].]

We have added a description of that study: “In their study, Protzko and colleagues [19] tested the prospective replicability of 16 novel empirical findings, using preregistration among other proposed current optimal practices. Here, 86% of effects could be replicated (p < .05) and effect sizes were 97% that of the original studies.”.

R: perhaps worth writing that these were randomly sampled for the literature.

[L88: Hardwicke et al. [25] found that only 3% of 188 examined articles from 2014 to 2017 included a preregistration statement]

We have changed the sentence accordingly: “Hardwicke et al. [25] found that only 3% of 188 examined articles from 2014 to 2017 which were randomly sampled from the literature included a preregistration statement...”.

R: can you list these other open science practices somewhere. Many people have their own interpretation of what OS include...and sometimes it’s unfortunately only open access.

[L95: Additionally, Logg and Dorison [27] showed that preregistration lags behind other open science practices]

We have included information about what these other techniques are: “...preregistration lags behind other open science practices such as open data and open material...”.

R: typo?

[L101f: Others worry that someone might take their preregistered and thus, publicly available, study idea and publish it before them]

We have removed the commas from this sentence.

R: Is this one of your hypotheses? Wouldn’t you need to collect data on whether they preregistered in the future to answer it? Consider rewording so it doesn’t appear as a hypotheiss.

[L155ff: The theory of planned behavior [40,41] can be applied to the context of preregistration to significantly predict researchers’ intention to preregister their studies in the near future, using a moderated multiple regression model]

This is the framework of our hypotheses, which we enumerate below. Since we did not investigate the actual behavior of the participants, we refer to “researchers’ intention to preregister” (not them actually preregistering) here.

R: unclear what this means.

[L229: Scales were recoded from “1 to 7” to “-3 to +3” for data analyses yielding a middle category which has absolute meaning.]

We have added: “(i.e., 0 = neutral opinion, neither agreement nor disagreement)”.

R: The questions were displayed in random order? Or the options to choose within each question were presented in random order. Clarify.

[L237f: Whenever applicable, options were displayed in randomized order to eliminate potential sequence effects]

We added: “response options”.

R: can you include that definition here.

[L248f: Before any items related to preregistration were shown, a definition of preregistration was presented...]

Since the definition is quite long, we have instead included references to the supporting information where it is displayed.

R: how? What was “correct” understanding?

[L249: and correct understanding was checked]

For this check, parts of the definition (e.g., “Preregistrations are assigned a time stamp.”) were presented alongside incorrect information (e.g., “Preregistration is typically done after data analysis.”) and participants had to select the correct options.

R: mean or median? I'd suggest reporting the median and interquartile range.

[L254: It took participants about 18 minutes (SD = 8 min, range = 44 min) on average to complete the survey...]

Since the difference between median and mean is not as important here than for other variables, we would like to adhere to reporting the mean, standard deviation, and range as preregistered. However, if you would like to take a look at the median and IQR, here are the requested values: Median = 16.37 min, IQR = 9.87 min.

R: how?

[L255: (times were adjusted for interruptions)]

A description was added: “... (times were adjusted for interruptions by replacing the completion time of pages that took participants more than two hours or 3 x SD of a normal distribution, with the page median of the other participants).”.

R: I imagine the ethics apprval ID is somewhere else in the manuscript. If not, I'd recommend putting it here.

[L257f: The survey was approved by the ethics committee of Trier University, Germany.]

We added more specific information about the ethical approval.: “The survey was approved by the ethics committee of Trier University, Germany (approval number: 07/2020). This form of consent was obtained in writing.”.

R: I recommend using the active voice to improve clarity. For example, by saying “we conducted..,” you remove the abiguity about whether it was the same team who did the pilot and full study.

[L261: A pilot study was conducted]

We changed the sentence accordingly: “We conducted a pilot study ...”.

R: unclear who participated and who was screened out. Consider rewording.

[L270: In this time, 29 participants (17 PhD students, three postdocs, seven professors, and two members of other academic groups which were screened out) started the survey...]

We changed the sentence accordingly: “(17 PhD students, three postdocs, seven professors, and two members of other academic groups; the latter were screened out)”.

R: but a much lower response rate for certain groups.

[L271f: yielding an overall response rate of 10%.]

We cannot provide any precise information on the response rate of individual academic groups here, since it is not clear how many persons of each group were contacted (since for our recruitment strategy we did not determine in advance who belonged to which academic group but invited all authors of the screened articles). Additionally, we only used the pilot study’s overall response rate for informing our main study recruitment strategy. Thus, we would like to refrain from adding the suggested statement in the manuscript.

R: how was this tested? Didn’t all the postdocs not complete the survey? What do you mean by “dropping out”?

[L272f: No specific patterns in dropout behavior were found (e.g., dropping out at a certain position).]

For inspecting the dropout behavior, we mainly looked at whether there was a particular point (page) in the survey where a particularly large number of people dropped out, which was not the case. We agree that it is somewhat striking that all postdocs dropped out, but the number of participants in the pilot study was generally very small. In the PhD student group, it was also the case that three people dropped out, but since here more participants had started the study, there were still 14 PhD students left who completed the study to the end. Due to the small number of participants in the pilot, we therefore did not take any further measures.

R: L303: How were the effect sizes chosen for the power analyses?

We provide this information below: “As comparable effect sizes, R² based on the averaged correlations of individual variables were searched for in the aforementioned meta-analyses, and the smallest ones were chosen for each power analysis.”

R: Why did you choose this effect size as the minimal effect size of interest? Do you have a justification for why this effect size, and not a smaller one, would also be interesting?

[L315: The percentage of variance of intention that was explained by attitudes, subjective norm and perceived behavioral control combined, ranged between 30.4% < R² < 44.3% [43,45–48].]

For our power analysis, we looked at effect sizes from several meta-analyses which investigated the theory of planned behavior in different contexts. We assumed that the effect size for the prediction of intention would probably be similar in our context (i.e., regarding the intention to preregister in psychological researchers). Therefore, we used the smallest of the identified effect sizes from the meta-analyses in our power analyses.

R: What’s this?

[L344: habilitation]

 Habilitation is the highest university degree in many European countries.

R: Some groups were overrepresented. Consider rewording or explain why this overrepresentation was restricted to within a reasonable range.

[L350: This enabled us to reach the a priori computed power even if not all quotas could be filled, while still ensuring that no group was overrepresented.]

We reworded this sentence: “This enabled us to reach the a priori computed power even if not all quotas could be filled, while still ensuring that the potential overrepresentation of individual groups remained within reasonable limits.”.

R: how? Did you just run two different analyses? If so, write so plainly.

[L378: As this may have introduced a sampling bias towards researchers who lean more positive towards preregistration, preregistration experience was controlled for in the statistical analyses...]

We added more information about how it was controlled: “...preregistration experience was controlled for in the statistical analyses aimed to draw inferences about the general population of psychological researchers by including it as a control variable in the hypotheses tests, and by providing descriptive reports separately for participants with and without preregistration experience.”.

R: no need for two decimal places.

[L386: 21.8% of participants were recruited through 386 PSYNDEX, 14.88% through OSF, 11.07% through Web of Science, 6.57% through PubMed, 6.23% through PsycInfo, 21.8% through email lists, and 17.65% through social media.]

We would like to keep the rounding to two decimal places here so that it is consistent everywhere.

R: typo?

[L391: The database, participants were recruited from, was considered in the analyses as described in the section descriptive reports]

We deleted both commas in this sentence.

R: I don’t understand this sentence. It makes it sound like everyone except 9 people responded faithfully.

[L438ff: Only data of participants that indicated having at least a bachelor’s degree in psychology, indicated faithful participation (n = 9 were excluded)…]

We edited this section slightly to facilitate comprehensibility: “At the end of the survey, participants were asked whether they responded faithfully. Here, nine participants indicated that their data should not be used in the analyses. Only data of participants that indicated having at least a bachelor’s degree in psychology, indicated faithful participation, and completed all pages were counted for quota fulfillment.”.

R: median or mean?

[L448: Mage = 34.99 years]

To ensure that the parameter shown is clearly understandable, we have written “Mean” in full everywhere.

R: the word “selected” may be less ambiguous here.

[L471: Additionally, multiple choice questions were recoded (originally: 1 = “not checked” and 2 = “checked”; new: 0 = “not checked” and 1 = “checked”)]

We have changed the wording accordingly.

R: p-values are fine, but it could also be useful to include the effect size and confidence intervals

[L530ff: The results showed that intention and perceived importance were similar between academic groups for participants with preregistration experience (both pcor = .952) but differed for participants that had not preregistered before (importance: F(3, 96) = 4.38, pcor = .018; intention: F(3, 63.35) = 4.94, pcor = .015), as displayed in Fig 3.]

We used a normal ANOVA for testing importance and a Brown-Forsythe test for intention. The confidence intervals given for these tests refer to something different respectively (ANOVA: confidence interval for each factor level; Brown-Forsythe: weighted sum of the square of the difference between the true means and the weighted average of the true means). Also, comparable effect sizes cannot be reported for both procedures because they also differ between procedures. To avoid potential misinterpretation, we would therefore like to refrain from reporting confidence intervals/effect sizes here.

R: clarify that this is the mean. It would also be useful to know the median and interquartile range.

[L544: having preregistered 6.05 studies on average]

We have clarified that the displayed parameter is the mean and have added the median and IQR: “having preregistered six studies on average (Mean = 6.05, Median = 3, SD = 10.68, IQR = 3, range = 80).”.

R: these add to more than 100%. How can people select multiple options for where they “first” heard of prereg?

[L555: The majority of participants indicated that they first learned about preregistration in informal conversations with colleagues or peers (55.29% of 331 responses). Around a quarter of participants furthermore indicated hearing about preregistration for the first time at official events at the workplace (27.19%), lectures at university (25.38%), or from their supervisors (24.17%), and some participants also learned about it through projects at their university (19.03%).] 

This item is actually a multiple-choice item to cover the case of learning about preregistration from two sources at about the same time. We appreciate your general comment that the nature of the item (multiple choice, single choice, open text input item, etc.) was not clear enough, and have added this information throughout the descriptive reports.

R: again, over 100%. Explain if multiple options could be selected

[L561ff: Other common reasons included informal conversations with colleagues or peers (33.84%), or it being suggested by their supervisor (28.79%) or co-authors (20.71%). In some cases, participants also created their first preregistration because it was mandatory for a project (12.63%) or to get funding (4.55%).]

See above. For each item, we have indicated the item type.

R: unclear what this means

[L564: practicing courses]

We reworded this sentence: “Other reasons provided in the open text input section of this item were that it was considered useful or essential for publication (2.05%) and that it was part of courses that participants had attended (1.03%).”.

R: was this free text response or multiple choice? Could respondents select multiple options?

[L567f: Reported benefits of preregistration were related to transparency (37.89% of 190 responses), trustworthiness of science (37.37%), or reduced uncertainty (5.26%).]

See above. For each item, we have indicated the item type.

R: this writing is unclear to me.

[L608: Instead, researchers were more insecure what needed to be included in the preregistration...]

We changed the wording to increase clarity: “Instead, researchers were more insecure what aspects needed to be included in the preregistration...”.

R: influence how? Make them more likely to preregister?

[L635: Fig 5. Parties that influence participants’ decision ...]

In the second part of the figure heading, it becomes clear that we are referring to both the influence for and against preregistration: “Fig 5. Parties that influence participants’ decision for or against preregistration”.

R: I feel that every instance of “average” could be replaced with “mean” or “median” for clarity. In many cases, I feel median and interquartile range would be more informative.

[L642: On average...]

For all these cases, we have made it clear when we refer to the mean and have added information about the median and IQR.

R: it’s unclear to me why some of these questions have so few responses. Are these open-text box responses that were optional to respond? I recommend clarifying this throughout.

[L690: (23.08% of 39 responses)]

See below. We have added a description in the manuscript that the items used for the descriptive reports were optional.

R: unclear which Qs were multiple choice, radio button response, and freehand text. 

[P33]

See above. We have clarified the item type throughout the manuscript.

R: the manuscript is extremely thorough (which is great). But it makes it really hard to pull out the main interesting take home messages. 

[P35]

See above. We agree that the manuscript is very long, which may not make the main results immediately apparent. However, we have taken some measures to simplify this (e.g., sub-headings).

R: reword for clarity: “Research experience and preregistration experience”

[L791: Research and preregistration experience combined explained…]

We have edited this accordingly.

R: These are hard to interpret based on prereg experience, because I don’t know how much of the 100% has vs hasn’t preregistered before. 

[Fig 5 and Fig 6]

See above. We have added the total number of participants with/without preregistration experience in the footnotes of these figures to facilitate their interpretability.

R: a table of contents for the supplementary files would be helpful. Also, a “readme” file in for how to rerun the analyses could be helpful.

[Supporting information]

Thank you for the comment. An overview of the supporting Information files is attached below the article (see section supporting information after the references). In addition, we included detailed comments in the analysis scripts to increase usability.

R: These videos really help understand the survey. Would it be possible to also attach a pdf or word version of the survey?

[Supporting information: S6 and S7 Video]

We have added a PDF version of the questionnaire: “Additionally, the questionnaire is displayed in a PDF in S5 File.”

R: Table 1 in S13 clearly shows that where participants were recruited from has an impact on their responses. I’d like to see data for the random sample alone (possibly in the main manuscript, if not, at least in supplementary materials).

[Supporting information]

We have added a table containing information about all scale items, based on the data of the database sample, in S15. Also, we originally only excluded the OSF when calculating the proportion of participants that had already preregistered (see section “Proportion of participants with preregistration experience”), but based on your suggestion, we also wanted to provide this information for the database sample. Thus, we have added the following sentence in the manuscript: “When only the sample recruited through the databases is considered (thus excluding participants recruited via email lists and social media in addition to OSF Registries), the proportion of participants with preregistration experience remains almost unchanged (61.74% of 149 responses).”

R: clarify whether -3 is strongly agree or strongly disagree in this table.

[Supporting information: S15 Text]

We have added this information in the table notes.

R: This is the table I wanted to see much earlier. Perhaps in figure format. And particularly seperated into the random sample (i.e., not including OSF, mailing list, social media)

[Supporting information: S15 Text]

See below. We have added a table in the corresponding supplement (S15) with the information only for the random sample (i.e., participants recruited from the databases).

R: I find the table in S15 important. Perhaps the authors could consider moving it to the main manuscript. In particular, I feel another version of this table that presents only the results from the more random sample would be interesting (e.g., respondents identified by emails taken from databases – and not those from the OSF, mailing lists, or social media). This subset of the data could be compared to random samples in future research to see if things have changed. However, this comparison cannot be reasonably performed with the full sample, as this sample doesn’t represent a clear population (due to the sampling bias towards those recruited from the OSF, mailing lists, and social media who have a different attitude than a random sample—as evidenced in Table 1 of S13).

Thank you for this useful suggestion. We have added a table in the corresponding supplement (S15) with the information only for the random sample (i.e., participants recruited from the databases): “...mean, standard deviation, and distribution of responses were displayed per item in tables for both the general and the database sample (i.e., only including participants recruited via the databases) for easy inspection (see S15 Text).”.

R: In the descriptive results section it is unclear which questions were open-text response, multiple choice with one selection option, and multiple choice with multiple selections possible. I recommend clarifying this throughout.

Thank you for this important remark. We have clarified the item type throughout the manuscript.

R: The denominator for how many people responded to a question changes for each question, and in some cases is very small compared to the total sample (e.g., 32). As a reader, it’s not clear why some questions have so few responses. Could you clarify this?

Thank you for your inquiry. As described in the manuscript, we used both complete and incomplete datasets for the descriptive reports: “Since incomplete datasets were also used for these reports, parameters are based on all responses given to the respective item (N is provided in each case).” Additionally, the items used for the descriptive reports were not mandatory. Based on this circumstance, the number of responses differed between items. We have added the following description to the manuscript to make it clearer that these items were not mandatory: “Items which were relevant for the study procedure (e.g., informed consent, usability), as well as those important for the exclusions, quota sampling, and hypotheses tests were mandatory. Meanwhile, the items used for the descriptive reports were optional.”.

R: Congrats on the thorough paper, it must have been a large effort. I hope it gets the traction it deserves.

We are very grateful for your detailed feedback and look forward to publicly presenting the results of the Registered Report.

Reviewer #3

R: The reviewer wants to congratulate the authors on this outstanding article.

Having been a reviewer for the protocol/ stage 1, the hard work and rigor put into this manuscript is much appreciated.

Thank you for your feedback for both the stage 1 and stage 2 Registered Report, which has helped us a lot for improving the manuscript.

R: This article is close to being accepted but one comment from my side:

In the discussion section I am missing a point on how pre-registration is not to be treated as the “panacea” (as described by Hardwicke and Wagenmarkers in their pre-print https://osf.io/preprints/metaarxiv/d7bcu/ ). The limitation part of this article could benefit from discussing this point.

Thank you for your suggestion to add this point in the discussion, and for bringing this reference to our attention. which is a great piece of work and a really helpful resource. Following your comment, we have added the following paragraph to our manuscript (in the limitations section): “On a more general note, it should be borne in mind that that preregistration has a variety of benefits that help increase scientific rigor, but it is not a panacea (as argued by [87]). Therefore, preregistration should ideally be used together with complementary open science practices increasing transparency and confidence (see [87] for an overview). Even so, for comprehensive and long-term improvement of the credibility of scientific evidence, the academic system needs to be revised, for example, by incentivizing methodological quality instead of results.”.

---

## [Editor Report · Decision Letter 1]

17 Jan 2023

Registered Report: Survey on attitudes and experiences regarding preregistration in psychological research

PONE-D-22-30808R1

Dear Dr. Spitzer,

We’re pleased to inform you that your manuscript has been judged scientifically suitable for publication and will be formally accepted for publication once it meets all outstanding technical requirements.

Kind regards,

Florian Naudet, M.D., M.P.H., Ph.D.

Academic Editor

PLOS ONE
---

## [Editor Report · Acceptance letter]

6 Mar 2023

PONE-D-22-30808R1 

Registered Report: Survey on attitudes and experiences regarding preregistration in psychological research 

Dear Dr. Spitzer:

I'm pleased to inform you that your manuscript has been deemed suitable for publication in PLOS ONE. Congratulations! Your manuscript is now with our production department. 

Kind regards, 

on behalf of

Pr. Florian Naudet 

Academic Editor

PLOS ONE